# Graph Convolutional Kernel Machine versus Graph Convolutional Networks

**Zhihao Wu**[1], **Zhao Zhang**[1,2], **Jicong Fan**[1,3*]
[1]Shenzhen Research Institute of Big Data, Shenzhen, China
[2]Hefei University of Technology, Hefei, China
[3]The Chinese University of Hong Kong, Shenzhen, China
{zhihaowu1999,cszzhang}@gmail.com, fanjicong@cuhk.edu.cn

## Abstract

Graph convolutional networks (GCN) with one or two hidden layers have been widely used in handling graph data that are prevalent in various disciplines. Many studies showed that the gain of making GCNs deeper is tiny or even negative. This implies that the complexity of graph data is often limited and shallow models are often sufficient to extract expressive features for various tasks such as node classification. Therefore, in this work, we present a framework called graph convolutional kernel machine (GCKM)[1] for graph-based machine learning. GCKMs are built upon kernel functions integrated with graph convolution. An example is the graph convolutional kernel support vector machine (GCKSVM) for node classification, for which we analyze the generalization error bound and discuss the impact of the graph structure. Compared to GCNs, GCKMs require much less effort in architecture design, hyperparameter tuning, and optimization. More importantly, GCKMs are guaranteed to obtain globally optimal solutions and have strong generalization ability and high interpretability. GCKMs are composable, can be extended to large-scale data, and are applicable to various tasks (e.g., node or graph classification, clustering, feature extraction, dimensionality reduction). The numerical results on benchmark datasets show that, besides the aforementioned advantages, GCKMs have at least competitive accuracy compared to GCNs.

## 1 Introduction

Graph data are prevalent in science and engineering. In recent years, neural networks and deep learning methods have shown promising performance in handling various graph data. Graph Neural Networks (GNNs) have been developed into various basic frameworks, including famous GCN [Kipf and Welling, 2017], GIN [Xu *et al.*, 2019], GraphSAGE [Hamilton *et al.*, 2017], GAT [Velickovic *et al.*, 2018], etc. As one of the most attractive GNN paradigms, GCN, the graph convolutional network [Kipf and Welling, 2017], has gained widespread attention since it was proposed. Many widely developed domains like machine learning [Chen *et al.*, 2023b; Cai *et al.*, 2022; Fan *et al.*, 2022; Fan, 2022b], data mining [Chang *et al.*, 2021; Fan, 2021; Wu *et al.*, 2022], computer vision [Guan *et al.*, 2022; Cai and Fan, 2022; Wu *et al.*, 2023], etc., have also made extensive use of numerous GCN variants. Despite the better performance compared to classical graph methods (e.g., DeepWalk [Perozzi *et al.*, 2014], LINE [Tang *et al.*, 2015] and node2vector [Grover and Leskovec, 2016]), some recent studies have pointed out a few drawbacks of GCNs [Wang *et al.*, 2020; Bo *et al.*, 2021; Wang *et al.*, 2023]. Perhaps the failure of deep GCNs is the most notorious one that severely restricted the further development of GCNs.

---

*Corresponding author.
[1]The source code is available at `https://github.com/ZhihaoWu99/GCKM`.

37th Conference on Neural Information Processing Systems (NeurIPS 2023).

Containing both the self features and pair-wise relations of entities, graph is viewed as a more complex non-Euclidean data type, so there is a natural and wide desire to construct deeper GCNs. Unfortunately, the performance of GCN decreases when stacking more graph convolutional layers, which is defined as the so-called over-smoothing issue [Li *et al.*, 2018]. For instance, even a 7-layer GCN will inevitably generate indistinguishable node embeddings and lead to over 30% accuracy degradation [Liu *et al.*, 2020]. Recently, some methods modifying the basic structure of GCN have been proposed to handle the over-smoothing issue and make GCN deeper [Xu *et al.*, 2018; Klicpera *et al.*, 2019; Yang *et al.*, 2022]. Indeed, most of them have successfully alleviated the over-smoothing issue. But it seems counter-intuitive that these elaborate deep GCNs, which stack dozens or even hundreds of layers, can only laboriously maintain the performance that a 2-layer vanilla GCN achieves easily. We apply some GCNs to a node classification experiment on Cora following the settings of

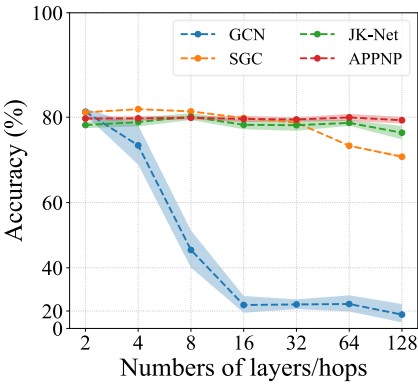

Figure 1: Accuracy and standard deviations of deep GCNs, SGC, and GCN with diverse layer/hop numbers.

the original GCN, as shown in Figure 1. APPNP [Klicpera *et al.*, 2019] and JKNet [Xu *et al.*, 2018] are two well-known models that attempted to pass information from shallow layers to the final output but benefited very limited from the deep neural networks. These explorations raise a question:

*Do we really need deep GCNs in common graph-based tasks?*

Actually, there have also been some efforts paid to simplifying GCN, showing comparative effectiveness to vanilla GCN, like SGC [Wu *et al.*, 2019], which removed nonlinearity and collapsed weight matrices to accelerate GCN. Impressively, it gained equal or even better performance on the benchmark graph datasets, which can partially be observed in Figure 1. Inspired by the success of SGC, some research moved further to decouple the graph convolution by individually performing an MLP (usually 2-layer) and several (can be hundreds) neighbor aggregations, called decoupled-style GCN [Nt and Maehara, 2019; Wang *et al.*, 2021]. All these works revealed that simple GCNs can achieve encouraging performance on commonly used graph datasets such as Cora, Citeseer, and Pubmed [Sen *et al.*, 2008]. In these datasets, the numerical node features and the adjacency matrices are already discriminative, which means further feature extractions via neural networks do not improve the performance significantly.

However, SGC removes the nonlinearities of GCN and essentially becomes a linear classifier in terms of features given by neighbor aggregations, which may limit the possibility of extending to more complex tasks. In general, considering the limitations of the deep GCNs and SGC, one can conclude that: ① *the neighbor aggregation is essential for GCNs;* ② *a simple feature mapping is helpful and sufficient but should be expressive enough.* These two points as well as the effectiveness of kernel methods in various machine learning problems [Cortes and Vapnik, 1995; Schölkopf *et al.*, 1998a; Fan and Chow, 2020; Fan *et al.*, 2022; Fan, 2022a] inspire us to establish a general and simple kernel-based framework, termed Graph Convolutional Kernel Machine (GCKM), for graph data. GCKM incorporates implicit feature mapping induced by kernels and neighbor aggregation over graphs, providing a new paradigm for graph-based machine learning. The main contributions of this work are summarized as follows:

- We propose a GCKM framework for graph-based learning. GCKM takes advantages of kernel learning and graph learning. Compared to GCNs, GCKMs have lower computational costs, higher interpretability, stronger theoretical guarantees, and stabler performances.
- We provide a generalization error bound for GCKM-based node classification and prove that graph structure can tighten this bound, both theoretically and empirically.
- We provide a few variants of GCKM that are useful in various graph-based learning problems such as node clustering and node embedding.
- We extend GCKMs to graph-level learning such as graph classification.

Comprehensive experiments demonstrate that the proposed GCKMs are as powerful as the state-of-the-art GCNs and significantly surpass several GCN-based methods on node semi-supervised classification and clustering tasks.

**Notations** Given an undirected attributed graph $\mathcal{G} = (\mathcal{V}, \mathcal{E})$, where $\mathcal{V}$ and $\mathcal{E}$ are the vertex set and edge set, respectively, and $|\mathcal{V}| = n$ is the number of nodes. The node features are represented as matrix $\mathbf{X} = [\mathbf{x}_1, \mathbf{x}_2, \ldots, \mathbf{x}_n]^\top \in \mathbb{R}^{n \times m}$, where $\mathbf{x}_i \in \mathbb{R}^m$ denotes the column feature vector of node $i$. And edges can be described by an adjacency matrix $\mathbf{A} \in \mathbb{R}^{n \times n}$, where $A_{ij} = 1$ if there exists an edge connecting node $i$ and $j$, otherwise $A_{ij} = 0$. We use $\mathbf{A}_i$ to denote the $i$-th row of $\mathbf{A}$. We denote the self-looped adjacency matrix as $\tilde{\mathbf{A}} = \mathbf{A} + \mathbf{I}$. The degree matrix corresponding to diagonal $\tilde{\mathbf{A}}$ is denoted by $\tilde{\mathbf{D}} \in \mathbb{R}^{n \times n}$, where $\tilde{D}_{ii} = \Sigma_{j=1}^n \tilde{A}_{ij}$. $\| \cdot \|$ denotes the Euclidean norm of vector. Let $\mathbf{y}_i \in \{0, 1\}^k$ be the label vector of node $i$ and $\mathcal{Y} = \{\mathbf{y}_i : \forall i \in \mathcal{L}\}$, where $\mathcal{L}$ denotes the index set of labeled nodes. For node classification, the task is to learn a model $f_\Theta(\cdot)$ from $\mathbf{X}$, $\mathbf{A}$, and $\mathcal{Y}$ to classify the unlabeled nodes.

## 2 Related Work

**Vanilla and Deep GCNs.** Kipf and Welling [2017] proposed GCN which further developed ChebyNet [Defferrard *et al.*, 2016] and formulated an efficient and effective graph convolutional operation. Up to now, numerous studies have tried to explore and interpret GCN, and a widely applied description of GCN decoupled the graph convolution into the following two steps

$$
\begin{aligned}
\bar{\mathbf{H}}^{(l+1)} &= \mathrm{AGGREGATE}\big(\{\mathcal{G}; \mathbf{H}^{(l)}\}\big) = \hat{\mathbf{A}}\mathbf{H}^{(l)}, \\
\mathbf{H}^{(l+1)} &= \mathrm{TRANSFORM}\big(\bar{\mathbf{H}}^{(l+1)}\big) = \sigma\big(\bar{\mathbf{H}}^{(l+1)}\mathbf{W}^{(l)}\big),
\end{aligned}
\tag{1}
$$

where $\mathbf{H}^{(l)}$ and $\mathbf{H}^{(l+1)}$ are the input and output representations of the $l$-th layer. $\hat{\mathbf{A}} = \tilde{\mathbf{D}}^{-\frac{1}{2}}\tilde{\mathbf{A}}\tilde{\mathbf{D}}^{-\frac{1}{2}}$ is the renormalized adjacency matrix and $\sigma(\cdot)$ is an activation function, e.g. $\mathrm{ReLU}(\cdot)$. These two operations compose the layer-wise propagation of GCN, but plenty of research has revealed that constructing a GCN with more such layers would deteriorate the performance. Li *et al.* [2018] first pointed out the over-smoothing issue is the key point of this phenomenon, that is, repeatedly multiplying the input node features by adjacency matrix eventually makes them more and more similar, resulting in indistinguishable node embeddings. This view has been broadly acknowledged, and many studies have been conducted on how to solve the over-smoothing issue. APPNP [Klicpera *et al.*, 2019] considered the relationship between GCN and PageRank to design a new aggregation scheme, and improve the performance of deep GCN. JKNet [Xu *et al.*, 2018] is another famous model for the same goal, which enabled each node to adaptively gain information from different hops of neighborhood. Nevertheless, different from the well-known deep CNNs, these approaches only alleviate the performance degradation but hardly benefit from the deep layers. More work on deep GCNs can be found in [Chen *et al.*, 2020; Sun *et al.*, 2021; Chen *et al.*, 2023a; Xu *et al.*, 2023].

**Simplified GCNs.** Although GCN put forward a simple formula of forward propagation rule, its mechanisms have not been fully discussed and may contain redundant computation, which suggests simplified GCNs. Among these, SGC [Wu *et al.*, 2019] removed the nonlinearity of GCN and only retained one learnable weight, so that built a faster model by precomputing the powers of the adjacency matrix. Wu *et al.* [2019] reported some interesting results that this intuitive way can still achieve the powerful expressive ability of vanilla GCN even with a relatively small solution space. Following this work, some decoupled-style GCNs have been designed, which decouple the two processes in (1) and then only stack the aggregation step while reducing the transformations. For example, Nt and Maehara [2019] claimed that graph structure only provided a denoising approach and GCN and SGC only performed low-pass filtering on node features. They then proposed gfNN to improve SGC from the perspective of graph signal processing. Liu *et al.* [2020] hypothesized that the coupling of these two operations causes not only more computations but also the over-smoothing issue, and introduced DAGNN by processing graph data via an MLP separately before performing the neighbor aggregation. In light of DAGNN, Dong *et al.* [2021] further analyzed the mechanism of decoupled GCNs, concluding the effectiveness of this kind of simplifying approaches.

## 3 Graph Convolutional Kernel Machine

### 3.1 General Framework

GCN can be decoupled as two key steps: neighbor aggregation and feature transformation, as formulated in (1). The former is a fixed operation and only the feature transformation contains the

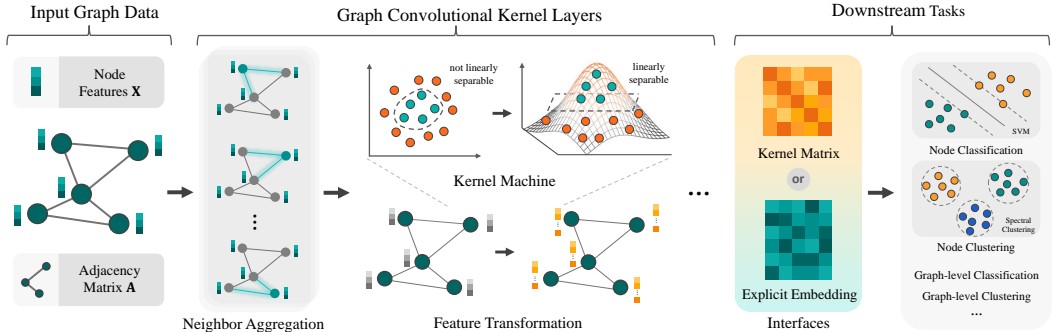

Figure 2: Overview of the proposed Graph Convolutional Kernel Machine (GCKM).

learnable weights $\mathbf{W}^{(l)}$, and it has been discovered that stacking these two operations significantly damaged the performance. Therefore, some research has been dedicated to solving this problem and making GCN deeper, but to little avail. On the other hand, various simplified GCNs reduced the learnable parameters and are competitive to those deeper GCNs, and SGC even simplified the feature transformation to one weight matrix without activation functions. In other words, despite the larger solution space provided by the nonlinearity and more learnable weights, improperly designed feature transformations may cause significant performance deterioration, while reduction of weights can lead to comparable power. Meanwhile, the representative SGC may be less expressive to cope with more complex data owing to its linear classifier. So it is crucial to present a light model by exploring an effective as well as efficient feature transformation approach.

Neighbor aggregation is commonly regarded as the pivotal component and makes the node representations more compact. Paired with a feature map to low-dimensional space, it may lead to indistinguishable node representations between different classes. Thus, to guarantee discriminative node representations, we propose to map node features to a higher-dimensional space via a transformation operation

$$\text{TRANSFORM}(\mathbf{X}) := \phi(\mathbf{X}) = [\phi(\mathbf{x}_1), \phi(\mathbf{x}_2), \ldots, \phi(\mathbf{x}_n)]^\top, \tag{2}$$

where $\phi : \mathbb{R}^m \to \mathbb{R}^M$ is a feature mapping and $M > m$. Explicitly designing a $\phi$ is usually costly especially when $M$ is very large or even infinity. Instead, we can take advantage of kernels that can induce $\phi$ implicitly. Let $k : \mathcal{X} \times \mathcal{X} \to \mathbb{R}$ be a kernel function, we have

$$k(\mathbf{x}, \mathbf{x}') = \langle \phi(\mathbf{x}), \phi(\mathbf{x}') \rangle = \phi(\mathbf{x})^\top \phi(\mathbf{x}'). \tag{3}$$

Thus we can obtain a kernel matrix of $\mathbf{X}$ as

$$\mathbf{K} = \begin{bmatrix} k(\mathbf{x}_1, \mathbf{x}_1) & \cdots & k(\mathbf{x}_1, \mathbf{x}_n) \\ \vdots & \ddots & \vdots \\ k(\mathbf{x}_n, \mathbf{x}_1) & \cdots & k(\mathbf{x}_n, \mathbf{x}_n) \end{bmatrix}. \tag{4}$$

There are many kernel functions available and each kernel function induces a specific feature map $\phi$. For instance, the $\phi$ induced by a polynomial kernel $k(\mathbf{x}, \mathbf{x}') = (\mathbf{x}^\top \mathbf{x}' + a)^b$ is a $b$-order polynomial feature map; the $\phi$ induced by a Gaussian kernel $k(\mathbf{x}, \mathbf{x}') = \exp\left(-\|\mathbf{x} - \mathbf{x}'\|^2/(2\sigma^2)\right)$ is an infinity-order polynomial feature map, namely, $M = \infty$.

We can alternatively perform the neighbor aggregation and the transformation (2), and obtain a layer-wise formulation of graph convolutional kernel machine as follows

$$\begin{aligned} \bar{\mathbf{H}}^{(l+1)} &= \text{AGGREGATE}\left(\{\mathcal{G}; \mathbf{H}^{(l)}\}\right) = \hat{\mathbf{A}}^q \mathbf{H}^{(l)}, \\ \mathbf{H}^{(l+1)} &= \text{TRANSFORM}\left(\bar{\mathbf{H}}^{(l+1)}\right) = \phi_{(l)}(\bar{\mathbf{H}}^{(l+1)}), \end{aligned} \tag{5}$$

where $\phi_{(l)}$ is the $l$-th implicit feature map induced by a kernel $k^{(l)}$ and the corresponding kernel matrix is $\mathbf{K}^{(l+1)} = \mathbf{H}^{(l+1)} \mathbf{H}^{(l+1)^\top}$. Here the $q$-th power of $\hat{\mathbf{A}}$ is adopted to allow several times graph convolution. Formulation (5) can be easily extended to multi-layer cases, but $\mathbf{H}^{(l+1)}$ is actually implicit and we calculate the kernel matrix $\mathbf{K}^{(l+1)}$ directly in practice. It is necessary to discuss

the connection between $\mathbf{K}^{(l+1)}$ and $\mathbf{K}^{(l)}$ and how to derive $\mathbf{K}^{(l+1)}$ layer by layer with only kernel matrices. Thus we will take the Gaussian kernel as an example, give the recursive formulation, and show how to derive the corresponding formulation in terms of kernel matrices. With a Gaussian kernel, we have $K_{ij}^{(l+1)} = \exp\left(-\frac{\Delta_{ij}^{(l+1)}}{2\sigma_{(l+1)}^2}\right)$, $(i,j) \in [n] \times [n]$, where $\Delta_{ij}^{(l+1)} = \left\|\bar{\mathbf{h}}_i^{(l+1)} - \bar{\mathbf{h}}_j^{(l+1)}\right\|^2$ is an element from $\mathbf{\Delta}^{(l+1)}$, which is a squared distance matrix of $\bar{\mathbf{H}}^{(l+1)}$. By a series of derivations (deferred to Appendix A), we have

$$\mathbf{\Delta}^{(l+1)} = \mathbf{d}_{\bar{\mathbf{K}}^{(l+1)}}\mathbf{1}_n^\top + \mathbf{1}_n\mathbf{d}_{\bar{\mathbf{K}}^{(l+1)}}^\top - 2\bar{\mathbf{K}}^{(l+1)}, \tag{6}$$

where $\bar{\mathbf{K}}^{(l+1)} = \hat{\mathbf{A}}^q\mathbf{K}^{(l)}(\hat{\mathbf{A}}^q)^\top$ and $\mathbf{d}_{\bar{\mathbf{K}}^{l+1}} = [\bar{K}_{11}^{(l+1)}, \bar{K}_{22}^{(l+1)}, \ldots, \bar{K}_{nn}^{(l+1)}]^\top$. Thus the recursive formulation can be summarized as

$$\begin{cases} \bar{\mathbf{K}}^{(l+1)} = \hat{\mathbf{A}}^q\mathbf{K}^{(l)}(\hat{\mathbf{A}}^q)^\top, \\ \mathbf{K}^{(l+1)} = \exp\left(-\frac{\mathbf{\Delta}^{(l+1)}}{2\sigma_{(l+1)}^2}\right). \end{cases} \tag{7}$$

Particularly, we define $\mathbf{K}^{(0)} = \mathbf{X}\mathbf{X}^\top$. Based on (7), GCKM with Gaussian kernels can be easily generalized to multi-layer cases. It is worth noting that similar compositions also apply to other kernels such as polynomial kernel and Laplace kernel, which will not be detailed here.

After obtaining the kernel matrix $\mathbf{K}^{(L)}$ from an $L$-layer GCKM, we can use Support Vector Machine (SVM) [Cortes and Vapnik, 1995] to perform node classification. Generally, SVM aims to solve

$$\min_{\mathbf{w},\boldsymbol{\xi}} \quad \frac{1}{2}\|\mathbf{w}\|^2 + \frac{\lambda}{n}\sum_{i=1}^n \xi_i \quad \text{s.t. } y_i\langle\varphi(\mathbf{x}_i),\mathbf{w}\rangle \geq 1 - \xi_i, \ \xi_i \geq 0, \ \forall i = 1,\ldots,n, \tag{8}$$

where $y_i \in \{+1,-1\}$ denotes the target label of training data $i$ and $\xi_i$ are the slack variables for the training data, and $\varphi$ is a feature map. The Lagrangian dual problem is

$$\max_{\mathbf{c}} \quad \sum_{i=1}^n c_i - \frac{1}{2}\sum_{i=1}^n\sum_{j=1}^n c_ic_jy_iy_j\varphi(\mathbf{x}_i)^\top\varphi(\mathbf{x}_j) \quad \text{s.t. } \sum_{i=1}^n c_iy_i = 0, \ 0 \leq c_i \leq \frac{\lambda}{n}. \tag{9}$$

Invoking $\mathbf{K}^{(L)}$ into (9), where $\varphi(\mathbf{x}_i) = \phi_{(L)}((\bar{\mathbf{h}}_i^{(L)})^\top)$, we obtain the following optimization problem of GCKM with $L$ layers for node classification:

$$\max_{\mathbf{c}} \quad \sum_{i=1}^n c_i - \frac{1}{2}\sum_{i=1}^n\sum_{j=1}^n c_ic_jy_iy_jK_{ij}^{(L)} \quad \text{s.t. } \sum_{i=1}^n c_iy_i = 0, \ 0 \leq c_i \leq \frac{\lambda}{n}. \tag{10}$$

For convenience, we call (10) graph convolutional kernel SVM (GCKSVM). Compared to GCN which is nonconvex, GCKSVM is convex and hence we can find its global optimal solution easily. In addition, kernel methods often have higher interpretability than neural networks and stronger generalization guarantees. Compared to SGC which is a linear classifier with respect to the final node representation, GCKSVM is a nonlinear classifier and hence is expected to perform better when the data are not linearly separable. Further justification and discussion will be presented in Figure 3 and its related explanations.

### 3.2 Generalization Bounds of GCKSVM for Node Classification

We analyze the generalization ability of GCKSVM for node classification. Different from [Bartlett and Shawe-Taylor, 1999; Bartlett and Mendelson, 2002; Grønlund *et al.*, 2020] that focuses on the primary problem of SVM, we focus on the dual problem, otherwise, the graph structure does not explicitly present in the bound.

Let $\hat{\mathbf{c}}$ be the solution to (10) and denote the corresponding $\mathbf{w}$ as $\hat{\mathbf{w}}$, where we use the Gaussian kernel for GCKSVM. Denote $S$ be the set of training data drawn from some distribution $\mathcal{D}$. Let $\mathcal{L}_S(\hat{\mathbf{c}}, \mathbf{K}^{(L)}) := \Pr_{(\varphi(\mathbf{x}),y)\sim S}[y\langle\varphi(\mathbf{x}),\hat{\mathbf{w}}\rangle \leq 1]$ be the training error and $\mathcal{L}_{\mathcal{D}}(\hat{\mathbf{c}}, \mathbf{K}^{(L)}) := \Pr_{(\varphi(\mathbf{x}),y)\sim\mathcal{D}}[\text{sign}(\langle\varphi(\mathbf{x}),\hat{\mathbf{w}}\rangle) \neq y] = \Pr_{(\varphi(\mathbf{x}),y)\sim\mathcal{D}}[y\langle\varphi(\mathbf{x}),\hat{\mathbf{w}}\rangle \leq 0]$ be the expected test error. We have the following generalization error bound.

**Theorem 1.** *Denote the index set of the support vectors as $\mathcal{V} = \{i : 1 \leq i \leq n, \ \hat{c}_i \neq 0\}$. For any $0 < \delta < 1$, it holds with probability at least $1 - \delta$ over a set of $n$ samples $S \sim \mathcal{D}$ that*

$$\mathcal{L}_{\mathcal{D}}(\hat{\mathbf{c}}, \mathbf{K}^{(L)}) \leq \mathcal{L}_S(\hat{\mathbf{c}}, \mathbf{K}^{(L)}) + O\left(\frac{\eta \ln n + \ln(1/\delta)}{n} + \sqrt{\frac{\eta \ln n + \ln(1/\delta)}{n} \cdot \mathcal{L}_S(\hat{\mathbf{c}}, \mathbf{K}^{(L)})}\right),$$

(11)

*where $\eta = \sum_{i \in \mathcal{V}} \hat{c}_i^2 + \sum_{i \in \mathcal{V}} \sum_{j \in \mathcal{V} \setminus i, y_i = y_j} \hat{c}_i \hat{c}_j K_{ij}^{(L)} - \sum_{i \in \mathcal{V}} \sum_{j \in \mathcal{V}, y_i \neq y_j} \hat{c}_i \hat{c}_j K_{ij}^{(L)} \leq \frac{\lambda^2 |\mathcal{V}|^2}{n^2}$.*

The proof for the theorem is in Appendix B. In the theorem, $K_{ij}^{(L)} = \exp\left(-\|(\hat{\mathbf{A}}^q)_i \mathbf{H}^{(L-1)} - (\hat{\mathbf{A}}^q)_j \mathbf{H}^{(L-1)}\|^2/(2\sigma_L^2)\right)$, which shows the direct connection between the error bound and graph. Empirically (see Appendix F.6), we find that given a fixed $\lambda$, the graph convolution reduces the number of support vectors $|\mathcal{V}|$ and has minor influence on the training error $\mathcal{L}_S(\hat{\mathbf{c}}, \mathbf{K}^{(L)})$, which eventually reduces the upper bound of test error. The fundamental reason is that incorporating the graph structure significantly improved the quality of the kernel matrix $\mathbf{K}^{(L)}$, in which the overall within-class similarity becomes much larger than the between-class similarity. In other words, the kernel matrix becomes more discriminative, which can be verified by Figure 4.

Note that the number of support vectors $|\mathcal{V}|$ in SVM is data-dependent and in order to theoretically show the influence of graph convolution on $|\mathcal{V}|$, we have to make the following assumption:

**Assumption 1.** *The aggregation step with graph $\mathcal{G}$ increases the inner product between the kernel feature maps of samples in the same class and reduces or does not change the inner product between the kernel feature maps of samples in different classes.*

This is a reasonable assumption because a useful graph should make the samples from different classes more distinguishable or at least make the samples from the same class more similar.

**Theorem 2.** *Given a graph $\mathcal{G}$ that satisfies Assumption 1, let $\varphi$ and $\varphi_{\mathcal{G}}$ be the kernel feature maps without and with aggregation on graph $\mathcal{G}$ respectively. The corresponding negative Lagrangian dual objectives (to minimize) are denoted as $\mathcal{L}(\mathbf{c}) := \frac{1}{2}\sum_{i=1}^n \sum_{j=1}^n c_i c_j q_{ij} - \sum_{i=1}^n c_i$, where $q_{ij} = y_i y_j \varphi(\mathbf{x}_i)^\top \varphi(\mathbf{x}_j)$ and $\mathcal{L}_{\mathcal{G}}(\mathbf{c}) := \frac{1}{2}\sum_{i=1}^n \sum_{j=1}^n c_i c_j q_{ij}^{\mathcal{G}} - \sum_{i=1}^n c_i$, where $q_{ij}^{\mathcal{G}} = y_i y_j \varphi_{\mathcal{G}}(\mathbf{x}_i)^\top \varphi_{\mathcal{G}}(\mathbf{x}_j)$. We have*

$$\mathcal{L}_{\mathcal{G}}(\mathbf{c}) \geq \mathcal{L}(\mathbf{c}) + \mathcal{R}(\mathbf{c}),$$

(12)

*where $\mathcal{R}(\mathbf{c}) \geq 0$ is a regularization term inducing sparsity in $\mathbf{c}$.*

The proof as well as the details about $\mathcal{R}(\mathbf{c})$ are in Appendix C. According to Theorem 2, the aggregation step introduces an additional sparse regularization term $\mathcal{R}(\mathbf{c})$, which will make $\mathbf{c}$ sparser, or in other words, reduce the number of support vectors $|\mathcal{V}|$. Considering both the theoretical and empirical results, we conclude that graph convolution leads to a tighter generalization error bound, which verifies the effectiveness of our GCKSVM. By the way, the interpretability of GCKSVM is higher than that of GCN, owing to the support vectors.

### 3.3 Extensions

Besides SVM, our method can be generalized to many other machine learning problems. We show a few examples of node-level learning in the following context.

- **GCKM for Spectral Clustering (GCKSC)** For node clustering, it is natural to perform spectral clustering [Ng *et al.*, 2001; Von Luxburg, 2007] using the kernel matrix $\mathbf{K}^{(L)}$. Instead of using the generally dense matrix $\mathbf{K}^{(L)}$, we retain only the largest values of each column to obtain a sparse matrix $\hat{\mathbf{K}}^{(L)}$, which often yields better clustering performance.

- **GCKM for Principal Component Analysis (GCKPCA)** It is straightforward to conduct Kernel PCA [Schölkopf *et al.*, 1998b] using $\mathbf{K}^{(L)}$, where the principal components can be used as the representations of the nodes of $\mathcal{G}$. Please refer to Appendix F.5 for experiments.

- We can also adapt GCKM to other learning models such as self-expressive models [Elhamifar and Vidal, 2013], Fisher linear discriminant analysis [Fisher, 1936], canonical correlation analysis [Hardoon *et al.*, 2004], etc.

Now we extend GCKM to graph-level learning. Suppose we have a set of graphs $\mathbb{G} = \{\mathcal{G}_1, \mathcal{G}_2, \ldots, \mathcal{G}_N\}$, where $|V_g| = n_g$, $g = 1, 2, \ldots, N$. For each $\mathcal{G}_g$, we use (7) to obtain $\mathbf{K}_g^{(L)}$ as well as the implicit node-level representations $\mathbf{H}_g^{(L)}$. For each $\mathcal{G}_g$, we compute the implicit graph-level representation as the summation of node-level representations, i.e.,

$$\mathbf{u}_g^{(L)} = (\mathbf{H}_g^{(L)})^\top \mathbf{1}_{n_g}, \quad g = 1, 2, \ldots, N. \tag{13}$$

Then for any two graphs $\mathcal{G}_g, \mathcal{G}_{g'}$, we have

$$\mathcal{K}_{gg'}^{(L)} := (\mathbf{u}_g^{(L)})^\top \mathbf{u}_{g'}^{(L)} = \mathbf{1}_{n_g}^\top \mathbf{H}_g^{(L)} (\mathbf{H}_{g'}^{(L)})^\top \mathbf{1}_{n_{g'}} = \mathbf{1}_{n_g}^\top \mathbf{K}_{gg'}^{(L)} \mathbf{1}_{n_{g'}}, \tag{14}$$

where $\mathbf{K}_{gg'}^{(L)} \in \mathbb{R}^{n_g \times n_{g'}}$ can be computed using (5) recursively. Consequently, we can obtain a graph-level kernel matrix as

$$\mathcal{K}_{\mathbb{G}}^{(L)} = \left[\mathcal{K}_{gg'}^{(L)}\right]_{(g,g') \in [N] \times [N]} = \begin{bmatrix} \mathbf{1}_{n_1}^\top & \cdots & \mathbf{0} \\ \vdots & \ddots & \vdots \\ \mathbf{0} & \cdots & \mathbf{1}_{n_N}^\top \end{bmatrix} \begin{bmatrix} \mathbf{K}_{11}^{(L)} & \cdots & \mathbf{K}_{1N}^{(L)} \\ \vdots & \ddots & \vdots \\ \mathbf{K}_{N1}^{(L)} & \cdots & \mathbf{K}_{NN}^{(L)} \end{bmatrix} \begin{bmatrix} \mathbf{1}_{n_1} & \cdots & \mathbf{0} \\ \vdots & \ddots & \vdots \\ \mathbf{0} & \cdots & \mathbf{1}_{n_N} \end{bmatrix} \tag{15}$$

Using (14) to compute $\mathcal{K}_{\mathbb{G}}^{(L)}$ has a high time complexity of $N(N-1)/2$ and using (15) has a high space complexity of $O((\sum_{g=1}^N n_g)^2)$. To reduce the cost, we can perform (15) on mini-batchs of $\mathbb{G}$. We have the following result (proved in Appendix D)

**Proposition 1.** *Suppose $k^{(L)}$ is a Gaussian kernel with $0 < \sigma_L < \infty$. Then $\mathcal{K}_{\mathbb{G}}^{(L)}$ is positive definite.*

$\mathcal{K}_{\mathbb{G}}^{(L)}$ can be applied to various graph-level learning tasks such as graph classification and clustering.

## 3.4 Fast and Explicit Feature Transformation for GCKM

GCKMs can be stacked as multi-layer model and applied to a wide range of tasks by calculating the kernel matrix, and has the advantages of being fast, explainable, and requiring few parameters. Despite this, the time and space complexities of GCKM are $O(n^2)$, which limits the application to very large graph datasets. To tackle this issue, we consider approximating the kernel by

$$k(\mathbf{x}, \mathbf{x}') = \langle \phi(\mathbf{x}), \phi(\mathbf{x}') \rangle \approx \langle \psi(\mathbf{x}), \psi(\mathbf{x}') \rangle, \tag{16}$$

where $\psi(\cdot)$ is a low-dimensional map using Random Fourier Feature [Rahimi and Recht, 2007]. The following theorem provides the key foundation of this approximation method:

**Theorem 3** (Bochner's theorem [Rudin, 1994]). *A continuous kernel $k(\mathbf{x}, \mathbf{x}') = k(\mathbf{x} - \mathbf{x}')$ on $\mathbb{R}^m$ is positive definite if and only if $k(\boldsymbol{\delta})$ is the Fourier transform of a non-negative measure.*

Specifically, for Gaussian kernel, let $p_{\text{RBF}}(\boldsymbol{\omega})$ be a probability distribution and also the Fourier transforamtion of $k_{\text{RBF}}(\boldsymbol{\delta})$. We first randomly sample $\{\boldsymbol{\omega}_1^{(l)}, \boldsymbol{\omega}_2^{(l)}, \ldots, \boldsymbol{\omega}_D^{(l)}\}$ from the probability distribution $p_{\text{RBF}}(\boldsymbol{\omega}) = \mathcal{N}\left(0, \frac{1}{\sigma}\mathbf{I}\right)$. Subsequently the multi-layer GCKM with explicit feature transformation, called GCKM-E, on the $i$-th sample is defined as $\mathbf{z}_i^{(l+1)} = \psi_{(l)}\left(\bar{\mathbf{z}}_i^{(l+1)}\right)$ where

$$\psi_{(l)}\left(\bar{\mathbf{z}}_i^{(l+1)}\right) = \sqrt{\frac{1}{D}} \left[\cos(\boldsymbol{\omega}_1^\top \bar{\mathbf{z}}_i^{(l+1)}), \ldots, \cos(\boldsymbol{\omega}_D^\top \bar{\mathbf{z}}_i^{(l+1)}), \sin(\boldsymbol{\omega}_1^\top \bar{\mathbf{z}}_i^{(l+1)}), \ldots, \sin(\boldsymbol{\omega}_D^\top \bar{\mathbf{z}}_i^{(l+1)})\right]^\top \tag{17}$$

and $\mathbf{z}_i^{(l+1)}$ and $\bar{\mathbf{z}}_i^{(l+1)}$ denote $i$-th columns vectors drawn from the transposes of $\mathbf{Z}^{(l+1)}$ and $\bar{\mathbf{Z}}^{(l+1)} = \hat{\mathbf{A}}^q \mathbf{Z}^{(l)}$, respectively. For detailed derivations, please refer to Appendix E. This approach allows GCKM to flexibly choose the dimensions of explicit node representations when $n$ is very large.

# 4   Experiment

In this section, we evaluate GCKMs on several real-world graph datasets. Comparison experiments of both semi-supervised node classification and node clustering show GCKMs' accuracy and efficiency. Due to the space limitation, some important experimental results are deferred to Appendix F.2 (more types of datasets and large-scale dataset), Appendix F.3 (over-smoothing problem), Appendix F.4 (GCKM with various kernels) and Appendix F.5 (GCKPCA visualization).

Table 1: Accuracy (mean% and standard deviation%) of all methods, note that the best results are highlighted in **orange** and the second-best results are highlighted in **blue**.

| | Standard Split | | | Random Split | | |
|---|---|---|---|---|---|---|
| | Cora | Citeseer | Pubmed | Cora | Citeseer | Pubmed |
| Chebyshev | 75.0 (0.7) | 63.8 (0.6) | 74.7 (0.8) | 76.1 (1.5) | 63.4 (3.8) | 75.4 (2.8) |
| GraphSAGE | 76.9 (0.5) | 63.3 (0.7) | 74.8 (0.2) | 76.3 (0.9) | 63.2 (1.1) | 73.9 (1.8) |
| GAT | 82.0 (0.5) | 69.8 (0.4) | 77.5 (0.2) | 79.2 (1.4) | 65.8 (2.9) | 78.1 (1.7) |
| GCN | 81.7 (0.7) | 70.5 (0.5) | 78.4 (0.4) | 80.6 (1.4) | 69.1 (1.8) | 77.4 (2.1) |
| SGC | 81.3 (0.0) | 68.4 (0.1) | 78.6 (0.1) | 79.6 (1.3) | 68.0 (1.7) | 76.6 (2.4) |
| APPNP | 79.5 (0.6) | 70.0 (0.8) | 78.8 (1.3) | 80.1 (1.8) | **69.2 (2.1)** | 77.7 (2.0) |
| JKNet | 77.9 (0.8) | 72.3 (0.1) | **80.1 (0.2)** | 78.0 (1.7) | 66.9 (2.3) | 77.0 (1.7) |
| DAGNN | **82.6 (1.1)** | 71.0 (0.5) | **80.0 (1.0)** | **81.8 (1.2)** | 68.4 (1.4) | 78.8 (1.4) |
| AdaGCN | 77.1 (0.1) | 69.4 (0.2) | 78.0 (0.1) | 76.7 (1.8) | 67.0 (1.2) | 77.3 (1.3) |
| AMGCN | 81.3 (0.4) | 70.4 (0.2) | 75.5 (0.9) | 80.9 (1.2) | 68.1 (1.4) | 74.2 (2.1) |
| DefGCN | 77.8 (1.0) | 67.5 (1.7) | 77.9 (0.6) | 78.4 (2.3) | 67.8 (2.6) | 77.1 (1.7) |
| GCKSVM-E | 82.4 (0.3) | **72.4 (0.4)** | 79.1 (0.4) | 80.8 (0.8) | 68.5 (1.3) | **79.2 (1.2)** |
| GCKSVM | **82.4 (0.0)** | **72.3 (0.0)** | 79.8 (0.0) | **83.3 (0.8)** | **71.9 (1.0)** | **80.9 (0.5)** |

**Datasets** We employ three most widely adopted citation networks Cora, Citeseer, and Pubmed for evaluations, they are formed as unweighted and undirected graphs where each node represents a paper and edges denote citations between papers. As for graph classification, IMDB-BINARY and IMDB-MULTI are movie collaboration datasets; COLLAB is a scientific collaboration dataset; MUTAG, PROTEINS, and PTC are three bioinformatics datasets. Details about these datasets and more datasets (e.g., social networks, paper networks and OGB-Arxiv) can be found in Appendix F.1.

**Compared Methods** For node classification, two types of GNNs are selected: Chebyshev [Defferrard *et al.*, 2016], GraphSAGE [Hamilton *et al.*, 2017], GAT [Velickovic *et al.*, 2018], GCN [Kipf and Welling, 2017] and SGC [Wu *et al.*, 2019] serve as classical baseline GNNs; APPNP [Klicpera *et al.*, 2019], JKNet [Xu *et al.*, 2018], DAGNN [Liu *et al.*, 2020], AdaGCN [Sun *et al.*, 2021], AMGCN[Wang *et al.*, 2020], DefGCN [Park *et al.*, 2022] are state-of-the-art models, in particular the former four ones are Deep GCNs. For node clustering, K-means and Spectral Clustering are two traditional baseline methods and GAE and VGAE [Kipf and Welling, 2016] are GCN-based baseline methods; ARGA, ARVGA [Pan *et al.*, 2018], DGI [Velickovic *et al.*, 2019], MVGRL [Hassani and Ahmadi, 2020], GALA [Park *et al.*, 2019], DFCN [Tu *et al.*, 2021] and S$^3$GC [Devvrit *et al.*, 2022] are all state-of-the-art GNN-based approaches. For graph classification, WL subtree kernel [Shervashidze *et al.*, 2011], AWL [Ivanov and Burnaev, 2018] are two baselines; DCNN [Atwood and Towsley, 2016], PATCHY-SAN [Niepert *et al.*, 2016] and DGCNN [Zhang *et al.*, 2018] are three deep learning methods; GCN, GraphSAGE and GIN [Xu *et al.*, 2019] are GNN-based methods.

**Experimental Settings** All the above methods are set as default following the original paper. We use a 2-layer GCKM for the experiments and the kernel is specified as Gaussian kernel, more detailed settings of GCKM can be found in Appendix F.1. In node classification, following vanilla GCN [Kipf and Welling, 2017], the nodes are split into three set: train set containing 20 samples per class, validation set and test set with 500 and 1,000 samples respectively, and the standard fixed split is same to [Yang *et al.*, 2016]. GCKM paired with kernel-based Spectral Clustering is leveraged for node clustering, which is described in Section 3.3. For graph classification task, we adopt the 10-fold cross validation following the settings of GIN Xu *et al.* [2019].

**Semi-supervised Node Classification** The results are reported in Table 1, where GCKSVM-E denotes GCKSVM with explicit features. Note that, we tune the hyperparameters of GCKSVM on the validation set and report the best performance. We have the following observations. **1)** In the case of standard split, GCKSVM achieves decent performance and beats several GNNs including deep GCNs, even though it is a shallow model. Besides, owing to that GCKSVM can obtain globally optimal solutions, the standard deviations are zero. **2)** In the case of random split, GCKSVM outperforms all GNNs on the three datasets and has the smallest standard deviations, showing higher stability. **3)** Deep GCNs are not substantially better than, or even worse than baselines, which further validates that deepening is not essential in common graph-based tasks. **4)** GCKSVM-E performs comparably to GCKSVM and other SOTA methods.

**Decision Boundary** To further explore the expressiveness, we illustrate the decision boundaries of SGC, APPNP, GCN, and the proposed GCKSVM in Figure 3. We generate a synthetic two-circle

Table 2: Graph classification accuracy (mean% and standard deviation%) of all methods, note that the best results are highlighted in **orange** and the second-best results are highlighted in **blue**.

| | IMDB-B | IMDB-M | COLLAB | MUTAG | PROTEINS | PTC |
|---|---|---|---|---|---|---|
| WL subtree | 73.8 (3.9) | 50.9 (3.8) | 78.9 (1.9) | **90.4 (5.7)** | 75.0 (3.1) | 59.9 (4.3) |
| DCNN | 49.1 | 33.5 | 52.1 | 67 | 61.3 | 56.6 |
| PATCHYSAN | 71.0 (2.2) | 45.2 (2.8) | 72.6 (2.2) | **92.6 (4.2)** | 75.9 (2.8) | 60.0 (4.8) |
| DGCNN | 70 | 47.8 | 73.7 | 85.8 | 75.5 | 58.6 |
| AWL | 74.5 (5.9) | 51.5 (3.6) | 73.9 (1.9) | 87.9 (9.8) | — | — |
| MLP | 73.7 (3.7) | 52.3 (3.1) | 79.2 (2.3) | 84.0 (6.1) | 76.0 (3.2) | **66.6 (6.9)** |
| GIN | **75.1 (5.1)** | **52.3 (2.8)** | **80.2 (1.9)** | 89.4 (5.6) | **76.2 (2.8)** | 64.6 (7.0) |
| GCN | 74.0 (3.4) | 51.9 (3.8) | 79.0 (1.8) | 85.6 (5.8) | **76.0 (3.2)** | 64.2 (4.3) |
| GraphSAGE | 72.3 (5.3) | 50.9 (2.2) | — | — | 75.9 (3.2) | 63.9 (7.7) |
| GCKSVM | **75.4 (2.4)** | **53.9 (2.8)** | **81.7 (1.5)** | 88.7 (7.6) | 74.5 (3.9) | **67.7 (5.4)** |

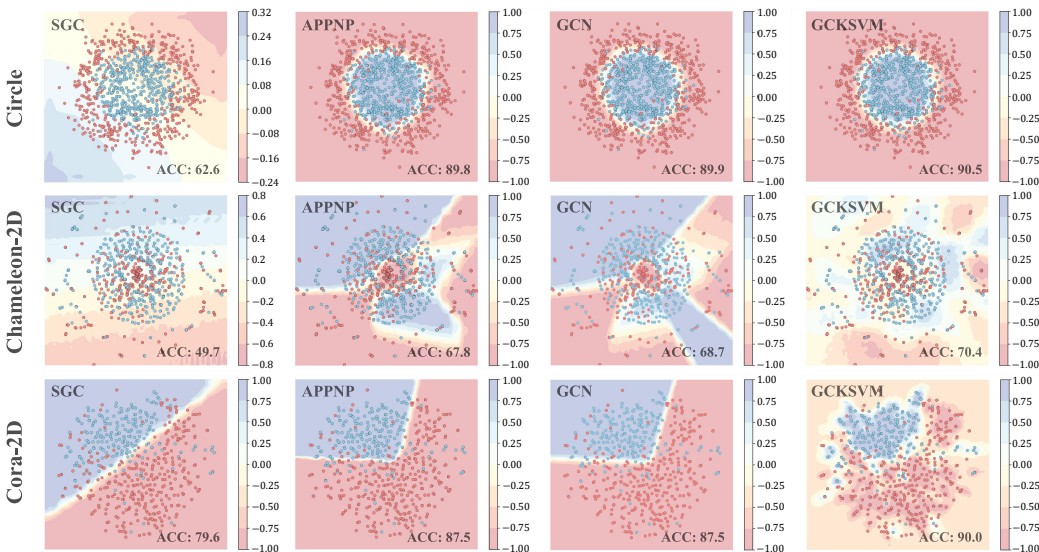

Figure 3: Decision boundary visualizations of SGC, APPNP, GCN, and GCKM on Circle (row 1), Chameleon-2D (row 2) and Cora-2D (row 3), where Circle is a synthetic dataset and Chameleon-2D and Cora-2D are two preprocessed real-world datasets.

dataset named Circle. We also consider two real datasets, denoted by Cora-2D and Chameleon-2D, where we select two classes of samples from the original datasets and map them to 2-D space using tSNE [Van der Maaten and Hinton, 2008]. We randomly labeled 200 samples for each dataset and constructed adjacency matrices by $k$-nearest neighbor algorithm, where the original graph structures are not used here. The details are described in Appendix F.1. Cora-2D is the simplest case where the decision boundary is easy to construct, Circle is with a clear but non-linear decision boundary, and Chameleon-2D is the most complex one.

We have the following conclusions. **1)** Consistent with our previous discussion, the nonlinearity provided by SGC is low even if we increase the power of $\hat{A}$ to a large value such as 50. Therefore, SGC performs poorly on Circle and Chameleon-2D, where its decision boundaries are approximately linear. This is unexpected considering its performance in the previous experiment. An explanation is that these commonly used graph datasets can be well classified linearly in high-dimension spaces, rather than these 2D spaces. **2)** The deep model APPNP fits these datasets with similar boundaries as GCN, either simple or complex, revealing that existing deep GCNs may not provide more powerful expressiveness. **3)** GCKSVM gains advantages over the three other models, whose decision boundaries fit these datasets well, attributed to which GCKM explicitly maps the samples to high-dimensional space to achieve better separability.

**Graph Classification** The results of graph classification tasks are recorded in Table 2. Compared to both classical baselines and GNNs, especially GIN designed for Graph-level tasks, GCKSVM shows

Table 3: ACC, NMI, and ARI of all methods, note that the best results are highlighted in **orange** and the second-best results are highlighted in **blue**.

| | Cora | | | Citeseer | | | Pubmed | | |
|---|---|---|---|---|---|---|---|---|---|
| | ACC | NMI | ARI | ACC | NMI | ARI | ACC | NMI | ARI |
| K-means | 49.20 | 32.10 | 22.90 | 54.00 | 30.50 | 27.80 | 59.50 | 31.50 | 28.10 |
| SC | 36.70 | 12.60 | 3.10 | 23.80 | 5.50 | 1.00 | 52.80 | 9.70 | 6.20 |
| GAE | 59.60 | 42.90 | 34.70 | 40.80 | 17.60 | 12.40 | 67.20 | 27.70 | 27.90 |
| VGAE | 50.20 | 32.90 | 25.40 | 46.70 | 26.00 | 20.50 | 63.00 | 22.90 | 21.30 |
| ARGA | 64.00 | 44.90 | 35.20 | 35.20 | 35.00 | 34.10 | 66.80 | 30.50 | 29.50 |
| ARVGA | 64.00 | 45.00 | 37.40 | 54.40 | 26.10 | 24.50 | 69.00 | 29.00 | 30.60 |
| DGI | 55.40 | 41.10 | 32.70 | 51.40 | 31.50 | 32.60 | 58.90 | 27.70 | 31.50 |
| MVGRL | 73.20 | 56.20 | 51.90 | 68.10 | 43.20 | 43.40 | 69.30 | 34.40 | 32.30 |
| GALA | 74.59 | 57.67 | 53.15 | 69.32 | 44.11 | 44.60 | 69.39 | 32.73 | 32.14 |
| DFCN | 64.07 | 48.24 | 39.17 | 69.50 | 43.90 | 45.50 | 69.32 | 32.19 | 31.55 |
| S³GC | 74.20 | 58.80 | 54.40 | 68.80 | 44.10 | 44.80 | 71.30 | 33.30 | 34.50 |
| GCKSC | 74.30 | 54.95 | 52.89 | 71.27 | 43.57 | 46.51 | 71.31 | 32.24 | 34.21 |

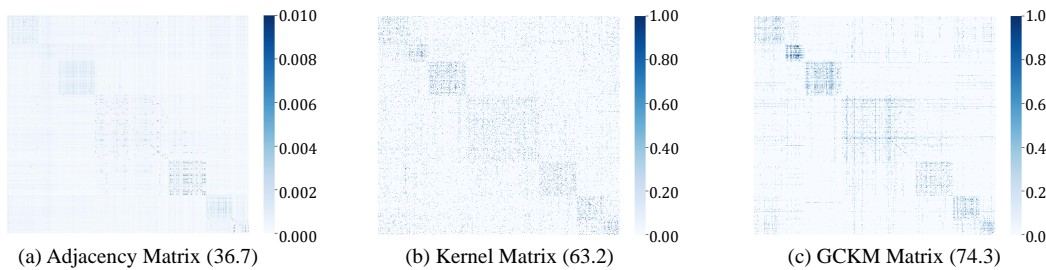

(a) Adjacency Matrix (36.7)    (b) Kernel Matrix (63.2)    (c) GCKM Matrix (74.3)

Figure 4: Visualizations of the adjacency matrix $\hat{\mathbf{A}}^q$, kernel matrix $[k(\mathbf{x}_i, \mathbf{x}_j)]_{i,j=1}^n$, and GCKM matrix $\mathbf{K}^{(2)}$ on Cora, with corresponding best-tuned clustering accuracy.

competitive power in graph classification. It is worth noting that GCKSVM outperforms the GNNs in the two relatively large datasets IMDB-M and COLLAB.

**Node Clustering** The comparison of all node clustering methods is shown in Table 3. Similar to the results of node classification, GCKM is well extended to node clustering tasks and is competitive with the deep-learning-based models. It is impressive that the vanilla GCN variants for clustering, GAE and VGAE both perform much worse than our GCKSC, demonstrating that GCKM can yield a discriminative kernel matrix. This can also be observed in Figure 4, where the powered adjacency matrix, the kernel matrix calculated on node features, and the kernel matrix of GCKM are visualized.

**Time Costs** Figure 5 shows the runtime of all methods on Pubmed. GCKSVM is much more efficient than all other methods except SGC.

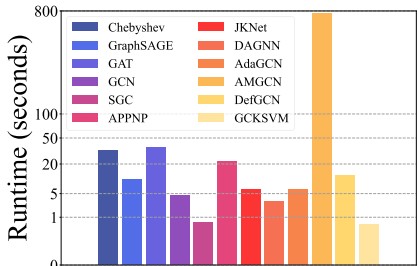

Figure 5: Runtimes of all methods.

## 5 Conclusions

After analyzing the limitations of GCN, SGC and deep GCNs, we proposed a framework of GCKM. GCKM integrated graph convolution with kernel learning and exhibited higher efficiency, interpretability, and stability. We extended GCKM to many graph-based machine-learning tasks, like node-level and graph-level classification and clustering. Experiments showed that GCKMs are, at least, as accurate as GCN and other SOTA GNNs. More importantly, compared to the representative simplified model SGC, GCKM is much more effective in handling non-linear data. One possible limitation of this work is that we did not systematically test other kernel functions, though we have found that the Gaussian kernel is better than the polynomial for GCKM.

## Acknowledgments

This work was partially supported by the National Natural Science Foundation of China under Grants No.62106211 and No.62072151, the General Program JCYJ20210324130208022 of Shenzhen Fundamental Research, the research funding T00120210002 of Shenzhen Research Institute of Big Data, the Guangdong Key Lab of Mathematical Foundations for Artificial Intelligence, Anhui Provincial Natural Science Fund for the Distinguished Young Scholars (2008085J30), Open Foundation of Yunnan Key Laboratory of Software Engineering (2023SE103), CCF-Baidu Open Fund and CAAI-Huawei MindSpore Open Fund, and the funding UDF01001770 of The Chinese University of Hong Kong, Shenzhen.

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

# A  Details about Recursive Formulations of GCKM

An $L$-layer GCKM on node feature $\mathbf{X}$ can be written as

$$\mathbf{H}^{(L)} = \phi_{(L)}\big(\hat{\mathbf{A}}^q \cdots \phi_{(l)}\big(\hat{\mathbf{A}}^q \cdots \phi_{(0)}\big(\hat{\mathbf{A}}^q \mathbf{X}\big) \cdots \big) \cdots \big). \tag{18}$$

Note that $\mathbf{H}^{(l+1)}$ is actually implicit and we directly obtain the kernel matrix $\mathbf{K}^{(l+1)} = \mathbf{H}^{(l+1)}\mathbf{H}^{(l+1)\top}$ in practice. Now we show how to derive $\mathbf{K}^{(l+1)}$ only by kernel matrix $\mathbf{K}^{(l)}$. Take the Gaussian kernel as an example, we have $K_{ij}^{(l+1)} = \exp\big(-\frac{\Delta_{ij}^{(l+1)}}{2\sigma_{l+1}^2}\big)$, $(i,j) \in [n] \times [n]$, where $\mathbf{\Delta}^{(l+1)}$ is a squared-distance matrix of the rows of $\bar{\mathbf{H}}^{(l+1)}$. To be more precise,

$$\Delta_{ij}^{(l+1)} = \|\bar{\mathbf{h}}_i^{(l+1)} - \bar{\mathbf{h}}_j^{(l+1)}\|^2 \tag{19}$$

$$= \bar{\mathbf{h}}_i^{(l+1)}\big(\bar{\mathbf{h}}_i^{(l+1)}\big)^\top - 2\bar{\mathbf{h}}_i^{(l+1)}\big(\bar{\mathbf{h}}_j^{(l+1)}\big)^\top + \bar{\mathbf{h}}_j^{(l+1)}\big(\bar{\mathbf{h}}_j^{(l+1)}\big)^\top, \tag{20}$$

where $\bar{\mathbf{h}}_i^{(l+1)}$ (a row vector) is the representation of the $i$-th node drawn from

$$\bar{\mathbf{H}}^{(l+1)} = \hat{\mathbf{A}}^q \mathbf{H}^{(l)} = \hat{\mathbf{A}}^q \phi_{(l-1)}\big(\bar{\mathbf{H}}^{(l)}\big). \tag{21}$$

Thus $\bar{\mathbf{h}}_i^{(l+1)} = (\hat{\mathbf{A}}^q)_i \phi_{(l-1)}\big(\bar{\mathbf{H}}^{(l)}\big)$, it follows from (19) that

$$\begin{aligned}
\Delta_{ij}^{(l+1)} &= \big(\hat{\mathbf{A}}^q\big)_i \phi_{(l-1)}\big(\bar{\mathbf{H}}^{(l)}\big)\phi_{(l-1)}^\top\big(\bar{\mathbf{H}}^{(l)}\big)\big(\hat{\mathbf{A}}^q\big)_i^\top - 2\big(\hat{\mathbf{A}}^q\big)_i \phi_{(l-1)}\big(\bar{\mathbf{H}}^{(l)}\big)\phi_{(l-1)}^\top\big(\bar{\mathbf{H}}^{(l)}\big)\big(\hat{\mathbf{A}}^q\big)_j^\top \\
&\quad + \big(\hat{\mathbf{A}}^q\big)_j \phi_{(l-1)}\big(\bar{\mathbf{H}}^{(1)}\big)\phi_{(l-1)}^\top\big(\bar{\mathbf{H}}^{(1)}\big)\big(\hat{\mathbf{A}}^q\big)_j^\top \\
&= \big(\hat{\mathbf{A}}^q\big)_i \mathbf{K}^{(l)}\big(\hat{\mathbf{A}}^q\big)_i^\top - 2\big(\hat{\mathbf{A}}^q\big)_i \mathbf{K}^{(l)}\big(\hat{\mathbf{A}}^q\big)_j^\top + \big(\hat{\mathbf{A}}^q\big)_j \mathbf{K}^{(l)}\big(\hat{\mathbf{A}}^q\big)_j^\top,
\end{aligned} \tag{22}$$

where $K_{ij}^{(l)} = \exp\big(-\frac{\|(\hat{\mathbf{A}}^q)_i \mathbf{H}^{(l-1)} - (\hat{\mathbf{A}}^q)_j \mathbf{H}^{(l-1)}\|^2}{2\sigma_l^2}\big)$, $(i,j) \in [n] \times [n]$. Particularly, we define $\mathbf{K}^{(0)} = \mathbf{X}\mathbf{X}^\top$. So $K_{ij}^{(l+1)}$ can be obtained through $\mathbf{K}^{(l)}$, i.e.,

$$K_{ij}^{(l+1)} = \exp\left(-\frac{(\hat{\mathbf{A}}^q)_i \mathbf{K}^{(l)}(\hat{\mathbf{A}}^q)_i^\top - 2(\hat{\mathbf{A}}^q)_i \mathbf{K}^{(l)}(\hat{\mathbf{A}}^q)_j^\top + (\hat{\mathbf{A}}^q)_j \mathbf{K}^{(l)}(\hat{\mathbf{A}}^q)_j^\top}{2\sigma_{l+1}^2}\right). \tag{23}$$

For convenience, letting $\bar{\mathbf{K}}^{(l+1)} = \hat{\mathbf{A}}^q \mathbf{K}^{(l)}(\hat{\mathbf{A}}^q)^\top$ and $\mathbf{d}_{\bar{\mathbf{K}}^{(l+1)}} = [\bar{K}_{11}^{(l+1)}, \bar{K}_{22}^{(l+1)}, \ldots, \bar{K}_{nn}^{(l+1)}]^\top$, we can calculate $\mathbf{K}^{(l+1)}$ in the following matrix form

$$\mathbf{K}^{(l+1)} = \exp\left(-\frac{\mathbf{d}_{\bar{\mathbf{K}}^{(l+1)}}\mathbf{1}_n^\top + \mathbf{1}_n \mathbf{d}_{\bar{\mathbf{K}}^{(l+1)}}^\top - 2\bar{\mathbf{K}}^{(l+1)}}{2\sigma_{l+1}^2}\right). \tag{24}$$

# B  Proof for Theorem 1

*Proof.* Since $\hat{\mathbf{c}}$ is the optimal solution of the dual problem, the optimal solution $\hat{\mathbf{w}}$ of the primary problem can be given by

$$\hat{\mathbf{w}} = \sum_{i \in \mathcal{V}} \hat{c}_i y_i \varphi(\mathbf{x}_i). \tag{25}$$

We have

$$\begin{aligned}
\|\hat{\mathbf{w}}\|_2^2 &= \left(\sum_{i \in \mathcal{V}} \hat{c}_i y_i \varphi(\mathbf{x}_i)\right)^\top \left(\sum_{i \in \mathcal{V}} \hat{c}_i y_i \varphi(\mathbf{x}_i)\right) \\
&= \sum_{i \in \mathcal{V}}\sum_{j \in \mathcal{V}} \hat{c}_i \hat{c}_j y_i y_j \varphi(\mathbf{x}_i)^\top \varphi(\mathbf{x}_j) \\
&= \sum_{i \in \mathcal{V}}\sum_{j \in \mathcal{V}} \hat{c}_i \hat{c}_j y_i y_j K_{ij}^{(L)} \\
&= \sum_{i \in \mathcal{V}} \hat{c}_i^2 K_{ii}^{(L)} + \sum_{i \in \mathcal{V}}\sum_{j \in \mathcal{V}\setminus i, y_i = y_j} \hat{c}_i \hat{c}_j K_{ij}^{(L)} - \sum_{i \in \mathcal{V}}\sum_{j \in \mathcal{V}, y_i \neq y_j} \hat{c}_i \hat{c}_j K_{ij}^{(L)} \\
&= \sum_{i \in \mathcal{V}} \hat{c}_i^2 + \sum_{i \in \mathcal{V}}\sum_{j \in \mathcal{V}\setminus i, y_i = y_j} \hat{c}_i \hat{c}_j K_{ij}^{(L)} - \sum_{i \in \mathcal{V}}\sum_{j \in \mathcal{V}, y_i \neq y_j} \hat{c}_i \hat{c}_j K_{ij}^{(L)}
\end{aligned} \tag{26}$$

On the other hand, we have

$$\|\hat{\mathbf{w}}\|_2^2 = \sum_{i \in \mathcal{V}} \sum_{j \in \mathcal{V}} \hat{c}_i \hat{c}_j y_i y_j K_{ij}^{(L)}$$
$$= \langle [\cdots \hat{c}_i y_i \cdots]^\top [\cdots \hat{c}_i y_i \cdots], [K_{ij}^{(L)}]_{i,j \in \mathcal{V}} \rangle$$
$$\leq \left\| [\cdots \hat{c}_i y_i \cdots]^\top [\cdots \hat{c}_i y_i \cdots] \right\|_F \left\| [K_{ij}^{(L)}]_{i,j \in \mathcal{V}} \right\|_F$$
$$= \left\| [K_{ij}^{(L)}]_{i,j \in \mathcal{V}} \right\|_F \sum_{i \in \mathcal{V}} \hat{c}_i^2 \tag{27}$$
$$\leq \left\| [K_{ij}^{(L)}]_{i,j \in \mathcal{V}} \right\|_F \frac{\lambda^2 |\mathcal{V}|}{n^2}$$
$$\leq \frac{\lambda^2 |\mathcal{V}|^2}{n^2}.$$

Now we use the following theorem, where the notations are a little different from ours.

**Theorem 3** ([Grønlund *et al.*, 2020]). *Let $d \in \mathbb{N}^+$ and let $R > 0$. Denote by $X$ the ball of radius $R$ in $\mathbb{R}^d$ and let $\mathcal{D}$ be any distribution over $X \times \{-1, 1\}$. For every $\delta > 0$, it holds with probability at least $1 - \delta$ over a set of $n$ samples $S \sim \mathcal{D}^n$, that for every $w \in \mathbb{R}^d$ with $\|w\|_2 \leq 1$ and every margin $\theta > 0$, we have*

$$\mathcal{L}_{\mathcal{D}}(w) \leq \mathcal{L}_S^\theta(w) + O\left( \frac{(R/\theta)^2 \ln n + \ln(1/\delta)}{n} + \sqrt{\frac{(R/\theta)^2 \ln n + \ln(1/\delta)}{n} \cdot \mathcal{L}_S^\theta(w)} \right)$$

According to our setting, we have $R = 1$ and $\theta = \frac{1}{\|\hat{\mathbf{w}}\|}$, since $\|\varphi(\mathbf{x}_i)\|^2 = K_{ii}^{(L)}$ and $y \langle \hat{\mathbf{w}}, \varphi(\mathbf{x}) \rangle \geq 1 \Leftrightarrow y \langle \frac{\hat{\mathbf{w}}}{\|\hat{\mathbf{w}}\|}, \varphi(\mathbf{x}) \rangle \geq \frac{1}{\|\hat{\mathbf{w}}\|}$. For convenience, let $\eta = \|\hat{\mathbf{w}}\|$, which was given by (27). Substituting these values into Theorem 3 and changing the notations, we obtain

$$\mathcal{L}_{\mathcal{D}}(\hat{\mathbf{c}}, \mathbf{K}^{(L)}) \leq \mathcal{L}_S(\hat{\mathbf{c}}, \mathbf{K}^{(L)}) + O\left( \frac{\eta \ln n + \ln(1/\delta)}{n} + \sqrt{\frac{\eta \ln n + \ln(1/\delta)}{n} \cdot \mathcal{L}_S(\hat{\mathbf{c}}, \mathbf{K}^{(L)})} \right).$$
$$\tag{28}$$

This finished the proof. □

## C  Proof for Theorem 2

*Proof.* Here we recall the Lagrangian dual problem:

$$\max_{\mathbf{c}} \quad \sum_{i=1}^n c_i - \frac{1}{2} \sum_{i=1}^n \sum_{j=1}^n c_i c_j y_i y_j \varphi(\mathbf{x}_i)^\top \varphi(\mathbf{x}_j) \quad \text{s.t.} \quad \sum_{i=1}^n c_i y_i = 0, \ 0 \leq c_i \leq \frac{\lambda}{n}. \tag{29}$$

For $\varphi$, we define

$$\mathcal{L}(\mathbf{c}) := \frac{1}{2} \sum_{i=1}^n \sum_{j=1}^n c_i c_j q_{ij} - \sum_{i=1}^n c_i, \ \text{where} \ q_{ij} = y_i y_j \varphi(\mathbf{x}_i)^\top \varphi(\mathbf{x}_j). \tag{30}$$

Similarly, for $\varphi_{\mathcal{G}}$, we let

$$\mathcal{L}_{\mathcal{G}}(\mathbf{c}) := \frac{1}{2} \sum_{i=1}^n \sum_{j=1}^n c_i c_j q_{ij}^{\mathcal{G}} - \sum_{i=1}^n c_i, \ \text{where} \ q_{ij}^{\mathcal{G}} = y_i y_j \varphi_{\mathcal{G}}(\mathbf{x}_i)^\top \varphi_{\mathcal{G}}(\mathbf{x}_j). \tag{31}$$

Then we need to minimize $\mathcal{L}(\mathbf{c})$ and $\mathcal{L}_{\mathcal{G}}(\mathbf{c})$ under the constraints $\sum_{i=1}^n c_i y_i = 0$, $0 \leq c_i \leq \frac{\lambda}{n}$. According to the Assumption 1, we have:

- if samples $i$ and $j$ are in the same class, $\varphi_{\mathcal{G}}(\mathbf{x}_i)^\top \varphi_{\mathcal{G}}(\mathbf{x}_j) > \varphi(\mathbf{x}_i)^\top \varphi(\mathbf{x}_j)$ and $y_i y_j = 1$;

- if samples $i$ and $j$ are in different classes, $\varphi_{\mathcal{G}}(\mathbf{x}_i)^\top \varphi_{\mathcal{G}}(\mathbf{x}_j) \leq \varphi(\mathbf{x}_i)^\top \varphi(\mathbf{x}_j)$ and $y_i y_j = -1$.

Therefore, the following equality holds:

$$q_{ij}^{\mathcal{G}} = q_{ij} + \epsilon_{ij}, \quad \epsilon_{ij} \geq 0, \ \forall (i,j) \in [n] \times [n] \tag{32}$$

For convenience, let $\bar{\epsilon}_i = \min_j \epsilon_{ij}$ and $\tilde{\epsilon} = \min_i \bar{\epsilon}_i$. We have

$$
\begin{aligned}
\mathcal{L}_{\mathcal{G}}(\mathbf{c}) :=& \frac{1}{2} \sum_{i=1}^{n} \sum_{j=1}^{n} c_i c_j (q_{ij} + \epsilon_{ij}) - \sum_{i=1}^{n} c_i \\
\geq& \frac{1}{2} \sum_{i=1}^{n} \sum_{j=1}^{n} c_i c_j q_{ij} + \frac{1}{2} \sum_{i=1}^{n} \bar{\epsilon}_i \sum_{j=1}^{n} c_i c_j - \sum_{i=1}^{n} c_i \\
=& \mathcal{L}(\mathbf{c}) + \frac{1}{2} \sum_{i=1}^{n} \tilde{\epsilon}_i c_i \sum_{j=1}^{n} c_j \\
\geq& \mathcal{L}(\mathbf{c}) + \frac{1}{2} \left( \tilde{\epsilon} \|\mathbf{c}\|_2^2 + \sum_{i=1}^{n} \bar{\epsilon}_i c_i \|\mathbf{c}_{/i}\|_1 \right),
\end{aligned}
\tag{33}
$$

where $\mathbf{c}_{/i} = [c_1, \ldots, c_{i-1}, c_{i+1}, \ldots, c_n]^\top$. It is known that the $\ell_1$-norm $\|\cdot\|_1$ is a convex relaxation of the $\ell_0$-norm $\|\cdot\|_0$, i.e., the number of nonzero elements in a vector. Denote $\mathcal{R}(\mathbf{c}) := \frac{1}{2}\big(\tilde{\epsilon}\|\mathbf{c}\|_2^2 + \sum_{i=1}^{n} \bar{\epsilon}_i c_i \|\mathbf{c}_{/i}\|_1\big)$. We see $\mathcal{R}(\mathbf{c})$ is very similar to the elastic net regularization and is able to induce sparsity. Actually, if we let $\kappa = \min_i \|\mathbf{c}_{/i}\|_1$, we have $\mathcal{R}(\mathbf{c}) \geq \frac{1}{2}\big(\tilde{\epsilon}\|\mathbf{c}\|_2^2 + \kappa \sum_{i=1}^{n} \bar{\epsilon}_i c_i\big) = \frac{1}{2}\big(\tilde{\epsilon}\|\mathbf{c}\|_2^2 + \kappa \|\mathrm{diag}(\bar{\epsilon})\mathbf{c}\|\big)$, where the second term is a weighted $\ell_1$-norm and also induces sparsity. $\square$

## D  Proof for Proposition 1

*Proof.* Let

$$
\mathbf{Q} = \begin{bmatrix} \mathbf{1}_{n_1} & \cdots & \mathbf{0} \\ \vdots & \ddots & \vdots \\ \mathbf{0} & \cdots & \mathbf{1}_{n_N} \end{bmatrix}, \quad
\bar{\mathbf{K}}^{(L)} = \begin{bmatrix} \mathbf{K}_{11}^{(L)} & \cdots & \mathbf{K}_{1N}^{(L)} \\ \vdots & \ddots & \vdots \\ \mathbf{K}_{N1}^{(L)} & \cdots & \mathbf{K}_{NN}^{(L)} \end{bmatrix}.
$$

It is easy to show that $\mathbf{Q}$ is full-rank. Because $k^{(L)}$ is a Gaussian kernel with $0 < \sigma_L < \infty$, $\bar{\mathbf{K}}^{(L)}$ is symmetric and always full-rank. It means $\bar{\mathbf{K}}^{(L)}$ can be factorized as $\bar{\mathbf{K}}^{(L)} = \mathbf{B}^\top \mathbf{B}$, where $\mathbf{B}$ is a square and full-rank matrix. Denote $\mathbf{P} = \mathbf{B}\mathbf{Q}$. It follows that

$$\mathcal{K}_{\mathbb{G}}^{(L)} = \mathbf{P}^\top \mathbf{P}. \tag{34}$$

Since $\mathbf{P}$ is full-rank, we conclude that $\mathcal{K}_{\mathbb{G}}^{(L)}$ is full-rank, which means $\mathcal{K}_{\mathbb{G}}^{(L)}$ is positive definite [Horn and Johnson, 2012]. $\square$

## E  Detailed Derivations of GCKM-E

**Theorem 2** (Bochner's theorem [Rudin, 1994]). *A continuous kernel $k(\mathbf{x}, \mathbf{x}') = k(\mathbf{x} - \mathbf{x}')$ on $\mathbb{R}^m$ is positive definite if and only if $k(\boldsymbol{\delta})$ is the Fourier transform of a non-negative measure.*

With positive definite and shift-invariant kernel $k(\cdot, \cdot)$ (e.g. Gaussian kernel), the Bochner's theorem points out that the kernel function can be expanded with harmonic basis as

$$k(\mathbf{x}_i - \mathbf{x}_j) = \int_{\mathbb{R}^d} p(\boldsymbol{\omega}) e^{j\boldsymbol{\omega}^\top (\mathbf{x}_i - \mathbf{x}_j)} d\boldsymbol{\omega} = \mathbb{E}_{\boldsymbol{\omega}} \big[ e^{j\boldsymbol{\omega}^\top \mathbf{x}_i} e^{-j\boldsymbol{\omega}^\top \mathbf{x}_j} \big], \tag{35}$$

where $p(\boldsymbol{\omega})$ is a probability distribution and also the Fourier transform of $k(\boldsymbol{\delta})$. For real-valued kernel functions, $e^{j\boldsymbol{\omega}^\top \mathbf{x}_i} e^{-j\boldsymbol{\omega}^\top \mathbf{x}_j}$ can be simplified by

$$e^{j\boldsymbol{\omega}^\top \mathbf{x}_i} e^{-j\boldsymbol{\omega}^\top \mathbf{x}_j} = \cos \boldsymbol{\omega}^\top (\mathbf{x}_i - \mathbf{x}_j) \tag{36}$$

according to Euler's formula. Then we can define function $\zeta_{\boldsymbol{\omega}}(\cdot)$ with $\zeta_{\boldsymbol{\omega}}(\mathbf{x}_i)\zeta_{\boldsymbol{\omega}}^\top(\mathbf{x}_j) = \cos\boldsymbol{\omega}^\top(\mathbf{x}_i - \mathbf{x}_j)$, so that it satisfies

$$\mathbb{E}_{\boldsymbol{\omega}}\left[\zeta_{\boldsymbol{\omega}}(\mathbf{x}_i)\zeta_{\boldsymbol{\omega}}^\top(\mathbf{x}_j)\right] = \mathbb{E}_{\boldsymbol{\omega}}\left[e^{j\boldsymbol{\omega}^\top\mathbf{x}_i}e^{-j\boldsymbol{\omega}^\top\mathbf{x}_j}\right] = k(\mathbf{x}_i - \mathbf{x}_j) \tag{37}$$

Here we simply adopt $\zeta_{\boldsymbol{\omega}}(\mathbf{x}) = \left[\cos(\boldsymbol{\omega}^\top\mathbf{x}), \sin(\boldsymbol{\omega}^\top\mathbf{x})\right]^\top$, and the kernel function can be approximated by combining several $\boldsymbol{\omega}$ drawn from distribution $p(\boldsymbol{\omega})$.

Specifically, for Gaussian kernel, we first randomly sample $\{\boldsymbol{\omega}_1, \boldsymbol{\omega}_2, \ldots, \boldsymbol{\omega}_D\}$ from the probability distribution $p_{\text{RBF}}(\boldsymbol{\omega}) = \mathcal{N}\left(0, \frac{1}{\sigma}\mathbf{I}\right)$. Subsequently the explicit feature transformation on the $i$-th sample $\mathbf{x}_i$ is defined as

$$\psi(\mathbf{x}_i) = \sqrt{\frac{1}{D}}\left[\cos(\boldsymbol{\omega}_1^\top\mathbf{x}_i), \ldots, \cos(\boldsymbol{\omega}_D^\top\mathbf{x}_i), \sin(\boldsymbol{\omega}_1^\top\mathbf{x}_i), \ldots, \sin(\boldsymbol{\omega}_D^\top\mathbf{x}_i)\right]^\top. \tag{38}$$

On the basis of the above derivations, the multi-layer GCKM with explicit feature transformation can be similarly defined:

$$\mathbf{Z}^{(l+1)} = \psi_{(l)}\left(\hat{\mathbf{A}}\mathbf{Z}^{(l)}\right), \tag{39}$$

where $\mathbf{Z}^{(l+1)}$ is the random Fourier features of input $\mathbf{Z}^{(l)}$. This approach allows GCKM to flexibly choose explicit node representations or kernel matrix as the interface, and the final representation can be low-dimensional and thus efficiently computed. By selecting a proper $D$, the convergence of this approximation and the expressive ability of the kernel can be simultaneously guaranteed.

# F Detailed Experimental Settings and Results

## F.1 Experimental Settings

Two types of graph datasets are employed in this paper. Each node-level dataset contains only a graph where each node is a sample, while each graph is regarded as a sample in the graph-level datasets. Here we list the key statistics of these datasets in Table 4 and 5 and provide detailed descriptions in the following.

**Node-level Datasets**

- Cora is a citation network containing a number of machine learning papers. Each node represents a paper and the edges represent the citation relationships between papers. Node features are the bag-of-words representations of papers, and all papers are divided into 7 categories according to their domains.

- Citeseer is also a well-known citation network with nodes, node features, and edges having the same meanings as Cora. Similarly, the papers are grouped into 6 classes.

- Pubmed is a citation network composed of $19,717$ scientific publications drawn from the PubMed database, which are classified into 3 categories and node features are also the bag-of-words representations.

- ACM is a paper network. Different from the above citation networks, the edges denote the co-author relationships between any two papers. Another difference is that the node features are bag-of-words representations of papers' keywords.

- CoraFull is a large version of the Cora dataset which has $19,793$ papers of 70 classes.

- Chameleon is a webpage network, in which nodes are Wikipedia pages of certain topics and edges are the page-page hyperlinks. Some informative nouns in the pages are extracted to be node features and the nodes are organized in 5 categories based on the average monthly traffic of the corresponding web pages.

- Actor is a subgraph of the film–director–actor–writer network which only includes actors as nodes. An edge between two actors exists when they occur on the same Wikipedia page.

- Squirrel is a webpage network as well, and all the settings are similar to the Chameleon dataset.

- UAI is a webpage network that has been used to test GCN for community detection. Nodes represent web pages and each edge represents a citation between two pages.

Table 4: Dataset statistics.

| Datasets | # Nodes | # Edges | # Classes | # Features | # Train/Val/Test | Data Type |
|---|---|---|---|---|---|---|
| Cora | 2,708 | 5,429 | 7 | 1,433 | 140/500/1,000 | Citation Network |
| Citeseer | 3,327 | 4,732 | 6 | 3,703 | 120/500/1,000 | Citation Network |
| Pubmed | 19,717 | 44,338 | 3 | 500 | 60/500/1,000 | Citation Network |
| ACM | 3,025 | 13,128 | 3 | 1,870 | 60/500/1,000 | Paper Network |
| Chameleon | 2,277 | 36,101 | 3 | 2,325 | 60/500/1,000 | Webpage Network |
| CoraFull | 19,793 | 65,311 | 70 | 8,710 | 1,400/500/1,000 | Citation Network |
| Actor | 7,600 | 15,009 | 5 | 932 | 100/500/1,000 | Social Network |
| Squirrel | 5,201 | 217,073 | 3 | 2,089 | 60/500/,1000 | Webpage Network |
| UAI | 3,067 | 28,311 | 19 | 4,973 | 367/500/,1000 | Webpage Network |

Table 5: Dataset statistics.

| | IMDB-B | IMDB-M | COLLAB | MUTAG | PROTEINS | PTC |
|---|---|---|---|---|---|---|
| # Nodes | 1,000 | 1,500 | 5,000 | 188 | 1,113 | 344 |
| # Classes | 2 | 3 | 3 | 2 | 2 | 2 |
| Avg # Nodes | 19.8 | 13.0 | 74.5 | 17.9 | 39.1 | 25.5 |
| Data type | Movie | Movie | Scientific | Bioinformatics | Bioinformatics | Bioinformatics |

- OGB-Arxiv dataset is a citation network with $169,343$ computer science arXiv papers, where each node is an arXiv paper and each edge indicates that a paper cites another paper. Each paper has a 128-dimensional feature vector, which is obtained by averaging embeddings of words in its title and abstract.

**Graph-level Datasets**

- IMDB-BINARY and IMDB-MULTI are movie collaboration datasets, where each graph represents an ego-network for each actor/actress. Each node corresponds to an actor/actress, and the edges represent the actors/actresses' co-occurrences in a movie. Graphs are classified according to the genre of the movies they come from.

- COLLAB is a scientific collaboration dataset, collected from three public collaboration datasets. Similar to the two datasets mentioned above, each graph is a self-network of different researchers, and these graphs are classified by the domains to which their corresponding researchers belong.

- MUTAG is a bioinformatics dataset. 188 mutagenic aromatic and heteroaromatic nitro compounds with 7 labels compose this dataset.

- PROTEINS is a bioinformatics dataset where nodes denote secondary structure elements with 3 labels. An edge between any two nodes means they are neighbors in the amino-acid sequence or 3D space.

- PTC is a bioinformatics dataset containing 344 chemical compounds that report carcinogenicity for male and female rats and it has 19 discrete labels.

For datasets ACM, Chameleon, CoraFull, Actor, Squirrel, and UAI, we randomly split them into the train, validation, and test sets and fix them for all the compared methods. For graph-level datasets split, we follow settings in [Xu *et al.*, 2019].

Then, we also briefly introduce the compared methods used in node-level tasks and graph-level tasks.

**Node-level Compared Methods**

- Chebyshev [Defferrard *et al.*, 2016] generalizes the convolution operation to the non-Euclidean space based on spectral graph theory, which leverages Chebyshev polynomial to design a localized graph convolutional filter.

- GraphSAGE [Hamilton *et al.*, 2017] is proposed to tackle the drawbacks of transductive GNNs. It trains the aggregation function from each node's local neighborhood, and this function can be performed on unseen nodes.

Table 6: Node classification accuracy (mean% and standard deviation%) of all methods, note that the best results are highlighted in **orange** and the second-best results are highlighted in **blue**. GraphSAGE encounters some errors on ACM and the corresponding result is marked as "—".

|  | ACM | Chameleon | CoraFull | Actor | Squirrel | UAI |
|---|---|---|---|---|---|---|
| Chebyshev | 82.8 (1.4) | 35.6 (0.2) | 57.2 (1.1) | 22.5 (0.4) | 23.0 (0.6) | 49.7 (0.4) |
| GraphSAGE | — | 44.0 (0.0) | 59.9 (0.7) | 21.0 (0.5) | 26.8 (0.2) | 41.7 (1.4) |
| GAT | 84.6 (0.5) | **51.1 (1.0)** | 62.4 (0.4) | 21.1 (0.7) | 28.4 (0.6) | 49.7 (3.0) |
| GCN | 88.8 (0.5) | 49.0 (1.8) | 62.8 (0.4) | 20.9 (0.9) | 30.9 (1.8) | 58.5 (1.1) |
| SGC | 80.8 (2.7) | 34.4 (1.1) | 62.9 (2.2) | 21.1 (1.3) | 23.3 (1.1) | 56.5 (3.5) |
| APPNP | 88.2 (0.0) | 50.4 (1.2) | 63.1 (0.5) | 22.6 (0.2) | 27.1 (0.1) | **62.3 (1.2)** |
| JKNet | 82.3 (0.6) | 50.3 (1.2) | 62.6 (0.0) | **30.4 (0.6)** | **37.2 (1.0)** | 45.6 (0.5) |
| DAGNN | 87.4 (0.9) | 33.9 (3.9) | **65.6 (0.3)** | 24.6 (1.7) | 18.7 (2.7) | 46.7 (12.4) |
| AdaGCN | 88.7 (0.0) | 43.8 (0.0) | **63.8 (0.0)** | 24.9 (0.0) | 25.7 (0.0) | 47.3 (1.1) |
| AMGCN | **90.4 (0.6)** | 34.2 (1.3) | 52.6 (0.7) | 26.8 (0.7) | 22.6 (1.3) | 59.3 (3.8) |
| DefGCN | 86.9 (0.6) | 49.5 (0.7) | 42.6 (2.3) | **31.3 (3.8)** | 28.4 (3.6) | 56.4 (2.1) |
| GCKSVM | **91.0 (0.0)** | **54.0 (0.0)** | 61.8 (0.0) | 29.5 (0.0) | **36.8 (0.0)** | **59.6 (0.0)** |

Table 7: Accuracy of several representative methods on large-scale dataset OGB-Arxiv, note that the best results are highlighted in **orange** and the second-best results are highlighted in **blue**.

| OGB-Arxiv | | | | | | | |
|---|---|---|---|---|---|---|---|
| # Nodes: 169,343 | | # Features: 128 | | # Edges: 1,166,243 | | # Classes: 40 | |
| Chebshev | GraphSAGE | GAT | GCN | SGC | APPNP | JKNet | GCKM-E |
| 69.7 (0.2) | 69.8 (0.2) | **70.2 (0.2)** | 69.5 (0.1) | 66.7 (0.0) | 69.3 (0.1) | 69.8 (0.2) | **71.0 (0.0)** |

- GAT [Velickovic *et al.*, 2018] is a classical spatial GNN equipping with self-attentional layers that adaptively learn different weights for edges.
- GCN [Kipf and Welling, 2017] is one of the most popular GNNs recently. It truncates the Chebyshev polynomial to first order and proposes a simple and effective graph convolution.
- SGC [Wu *et al.*, 2019] further simplifies GCN by removing the nonlinearity and collapsing the learnable weights, so as to build a faster model and achieve competitive performance.
- APPNP [Klicpera *et al.*, 2019] uses personalized Pagerank to improve vanilla GCN by solving the over-smoothing issue and derives a new framework with initial residual connections.
- JKNet [Xu *et al.*, 2018] establishes a jumping knowledge structure to combine representations from various layers to stack a deeper GNN.
- DAGNN [Xu *et al.*, 2023] decouples the neighbor aggregations and feature transformations to form a deeper GNN and further uses the attention mechanism to integrate the information from different depths.
- AdaGCN [Sun *et al.*, 2021] is also a deep GCN that considers the Adaboost strategy to fuse knowledge from distinct layers.
- AMGCN [Wang *et al.*, 2020] constructs a feature graph via $k$-NN algorithm and a multi-channel framework to enhance the vanilla GCN.
- DefGCN [Park *et al.*, 2022] improves the limitations of common GNNs with fixed graph convolution and has deformable graph convolution that allows nodes to adaptively capture long-range dependencies.

**Graph-level Compared Methods**

- WL subtree kernel [Shervashidze *et al.*, 2011] is a famous graph kernel, and AWL [Ivanov and Burnaev, 2018] is an approach for embedding entire graphs. Paired with SVM, they are selected as two baselines.
- DCNN [Atwood and Towsley, 2016], PATCHY-SAN [Niepert *et al.*, 2016] and DGCNN [Zhang *et al.*, 2018] are three deep learning methods for graph classification, and their performance is reported as in the original paper.

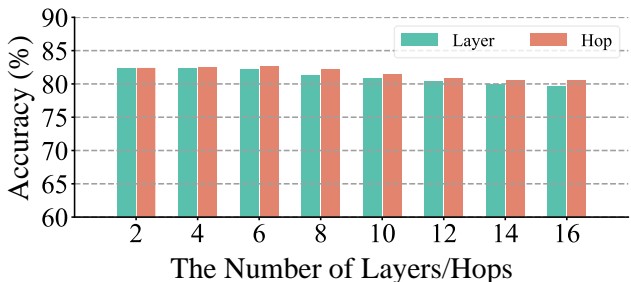

Figure 6: Accuracy of GCKSVM with varied layers/hops.

- GCN, GraphSAGE, and GIN [Xu *et al.*, 2019] serve as GNN-based compared methods, and they are all set following [Xu *et al.*, 2019]. For the methods having different variants, we report their best performance.

Note that the settings of these graph-level methods are the same as [Xu *et al.*, 2019].

**Settings about Figure 1.** We set the hidden dimensions as 32 for all hidden layers of GCN, APPNP, and JKNet, while SGC only has a learnable matrix $\mathbf{W} \in \mathbb{R}^{n \times c}$ where $n$ is the number of nodes and $c$ is the number of classes, and the learning rate is selected in $\{1 \times 10^{-2}, 1 \times 10^{-3}, 1 \times 10^{-4}\}$ and weight decay is selected in $\{5 \times 10^{-4}, 5 \times 10^{-5}, 5 \times 10^{-6}\}$.

**Settings of Boundary Visualization** For better illustration, we conduct a visualization experiment on the decision boundaries of SGC, APPNP, GCN, and GCKSVM. The purpose of this experiment is to explore how the three methods perform on linear, non-linear, and more complex data. So we select 2 classes of samples from Cora and Chameleon and map them to 2-D space by t-SNE. Besides, we generate a synthetic dataset (called Circle) consisting of noisy samples drawn from two concentric circles using Scikit-learn. For better boundary and performance, the numbers of layers are 2 and 8 for GCN and APPNP respectively, the hidden dimensions are 32, and the learning rate is set as $1 \times 10^{-3}$ and weight decay is $5 \times 10^{-5}$ for all models. SGC has only one learnable matrix and we set the power of adjacency as $50$ to show that the graph structure provide SGC with non-linearity but the non-linearity is very low.

### F.2 Supplementary Experiments of Node Classification

In this section, we evaluate GCKSVM on more types of graph datasets. As reported in Table 6, it can be seen that GCKSVM still has superior performance compared to the most state-of-the-art GNNs. Furthermore, we also evaluate the proposed GCKM on the OGB dataset Arxiv with over 160k nodes. Table 7 demonstrates that GCKM is still competitive with these GNNs on large-scale datasets. Although the complexity of GCKM is quadratic, we provide an efficient variant GCKM-E in Section 3.4 that leverages the random Fourier feature to explicitly derive a low-dimensional output instead of a kernel matrix. With this output, we can apply fast linear methods (e.g. linear SVM) for downstream tasks, which is still more efficient than GNNs needing a lengthy training process.

### F.3 Experiments and Discussions on Over-smoothing Issue

Thanks for your insightful comment. We have supplemented an experiment on this issue, which revealed that GCKM can be deeper and alleviate the over-smoothing issue. To be specific, two situations are considered in this experiment:

- GCKM with fixed 2 layers and varied hops of neighbors per aggregation ($q$ in Eq. (40))

$$\mathbf{H} = \phi_{(1)}(\hat{\mathbf{A}}^q \phi_{(0)}(\hat{\mathbf{A}}^q \mathbf{X})). \tag{40}$$

- GCKM with fixed 2 hops of neighbors per aggregation and varied layers

$$\mathbf{H} = \phi_{(L)}(\hat{\mathbf{A}}^2 \cdots \phi_{(0)}(\hat{\mathbf{A}}^2 \mathbf{X})). \tag{41}$$

From Figure 6, we have the following observations:

Table 8: Accuracy of GCKSVM with different kernel functions on Cora, Citeseer, and Pubmed.

| | Cora | Citeseer | Pubmed |
|---|---|---|---|
| Polynomial | 83.0 | 70.6 | 80.6 |
| Sigmoid | 79.7 | 69.2 | 75.6 |
| Laplacian | 82.2 | 70.5 | 79.0 |
| Gaussian | 82.4 | 72.3 | 79.8 |

- Deep GCKM performs more stably than deep GCN.
- GCKM's performance first improves then slightly decreases with increasing layers/hops
- Particularly, GCKM with fixed layers performs better and decreases less.

It is known that over-smoothing is caused by the aggregation step, that is, multiplying $\hat{\mathbf{A}}$ makes the representations of different nodes more and more indistinguishable. However, the analysis of over-smoothing does not consider activation functions and learnable weights, and it theoretically exists when the power of $\hat{\mathbf{A}}$ tends to infinity [Li *et al.*, 2018], which does not match the fact that GCN collapses with only 8 layers. A recent study of [Cong *et al.*, 2021] has pointed out that the over-smoothing problem might be an artifact of theoretical analysis and the failure of deep GNNs may not only be caused by the over-smoothing issue in the aggregation/message-passing step.

Although GCKM and GCN share a familiar aggregation step, the main difference between them is the transformation step, namely, GCN uses a linear layer to conduct explicit dimension reduction while GCKM employs implicit high-dimensional feature mapping. GCKM implicitly maps node features to a high-dimensional (even infinite) space after aggregation, and there may exist an appropriate space where node representation can be distinguished. In contrast, a recent study of [Guo *et al.*, 2023] has found that the node representations processed by deep GCN would collapse to be low-rank and lose expressive power. The appropriate space can be found by tuning the hyperparameters of GCKM, however, the number of hyperparameters will increase when building a deeper GCKM and it is time-consuming to search this space. That may be the reason why GCKM decreases slightly with too many layers and why fixing the number of layers improves the performance.

### F.4 Experiments on Various Kernels

To solve the limitations we stated in the conclusion section, we provided a complementary experiment on various kernel functions during the rebuttal process. Table 8 records the results corresponding to 2nd order polynomial kernel, sigmoid kernel, and Laplacian kernel. All the kernel functions show decent performance and the Gaussian kernel performs the best. Note that, this experiment is conducted with GCKSVM, under the settings of node classification with standard split.

### F.5 Experiments of GCKPCA

We added a visualization experiment for GCKPCA. To be specific, we first map the node features to 2-D space by PCA, Graph-regularized PCA (GPCA) [Jiang *et al.*, 2013] and GCKPCA (row 1), then further try to map them to 32-D space and leverage t-SNE to obtain the 2-D results (row 2). Figure 7 illustrates the comparison of mapping results of three methods. It can be observed that the subfigures of GCKPCA and GCKPCA + t-SNE both show the best separability between different classes, and GPCA performs slightly better than PCA.

### F.6 Empirical Experiments regarding Theorem 1

Theorem 1 demonstrated the connections between the generalization bound and graph structure, and experiments further provided the evidence. To verify the conclusion we have made from Theorem 1 in the main paper, we report the training error and numbers of support vectors in Table 9. In fact, we consider three cases:

- Non-graph-convolution: We replaced the affinity matrix $\hat{\mathbf{A}}$ in GCKM with an identity matrix $\mathbf{I}_n$, which means the graph structure is not used.

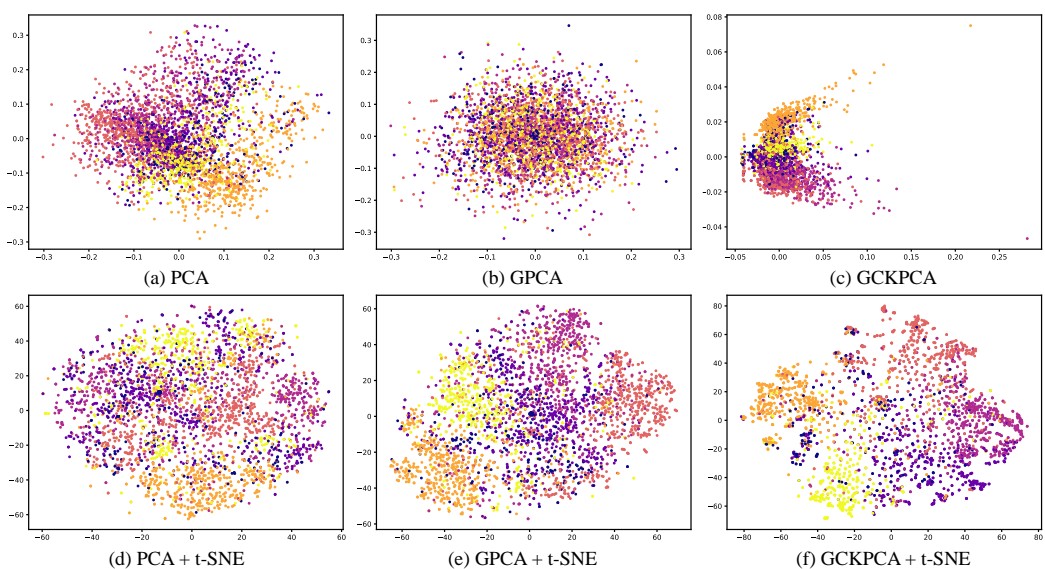

(a) PCA       (b) GPCA       (c) GCKPCA

(d) PCA + t-SNE       (e) GPCA + t-SNE       (f) GCKPCA + t-SNE

Figure 7: Visualization of PCA, GPCA and GCKPCA.

Table 9: The values of the training error (TE) and the number of support vectors $|\mathcal{V}|$. Note that $\lambda$ is the regularization parameter of SVM, $\mathbf{I}_n$ and $\mathbf{1}_n$ are $n \times n$ identity matrix and all-ones matrix respectively.

| | | $\mathbf{I}_n$ | | $\mathbf{1}_{n \times n} - \mathbf{I}_n$ | | $\hat{\mathbf{A}}^q$ | |
|---|---|---|---|---|---|---|---|
| | | TE | $|\mathcal{V}|$ | TE | $|\mathcal{V}|$ | TE | $|\mathcal{V}|$ |
| Cora | $\lambda = 1$ | 0.01 | 80 | 0.50 | 32 | 0.20 | 40 |
| | $\lambda = 10$ | 0.00 | 72 | 0.50 | 32 | 0.00 | 30 |
| | $\lambda = 50$ | 0.00 | 72 | 0.50 | 32 | 0.00 | 24 |
| | $\lambda = 100$ | 0.00 | 72 | 0.50 | 32 | 0.00 | 24 |
| | $\lambda = 1000$ | 0.00 | 72 | 0.50 | 32 | 0.00 | 24 |
| Citeseer | $\lambda = 1$ | 0.02 | 120 | 0.50 | 62 | 0.20 | 120 |
| | $\lambda = 10$ | 0.00 | 112 | 0.50 | 62 | 0.00 | 105 |
| | $\lambda = 50$ | 0.00 | 111 | 0.50 | 62 | 0.00 | 102 |
| | $\lambda = 100$ | 0.00 | 111 | 0.50 | 62 | 0.00 | 102 |
| | $\lambda = 1000$ | 0.00 | 111 | 0.50 | 62 | 0.00 | 102 |
| Pubmed | $\lambda = 1$ | 0.12 | 1206 | 0.48 | 802 | 0.18 | 1366 |
| | $\lambda = 10$ | 0.05 | 755 | 0.48 | 802 | 0.10 | 755 |
| | $\lambda = 50$ | 0.02 | 617 | 0.48 | 794 | 0.07 | 571 |
| | $\lambda = 100$ | 0.01 | 574 | 0.48 | 802 | 0.07 | 508 |
| | $\lambda = 1000$ | 0.00 | 548 | 0.48 | 804 | 0.11 | 396 |

- Strongly connected graph: $\hat{\mathbf{A}}$ in GCKM is replaced with $\mathbf{I}_n - \mathbf{1}_{n \times n}$, which means every node is connected with all other nodes.
- The normal GCKM.

We see that with fixed $\lambda$, graph convolution significantly reduces the number of support vectors, which verifies the correctness of Theorem 2. Thus, according to Theorem 1, the upper bound (monotonically increasing with $|\mathcal{V}|$) of test error can be tighter compared to the one using a non-graph-convolution kernel, which means that the test error is potentially smaller. So we can conclude that the graph structure significantly improves the quality of the kernel matrix.

