# OpenReview forum: "Graph Convolutional Kernel Machine versus Graph Convolutional Networks"
_NeurIPS.cc/2023/Conference — NeurIPS 2023 poster_

### Official Review · Reviewer_PZjL · 2023-07-03

**Soundness:** 3 good
**Presentation:** 3 good
**Contribution:** 3 good
**Rating:** 7
**Confidence:** 5

**Summary:**

The paper presents a framework called graph convolutional kernel machine (GCKM) for graph-based machine learning. GCKMs are built upon kernel functions integrated with graph convolution. Within the framework of GCKM, the authors propose GCKSVM for node classification, GCKSC for node clustering, and extensions for graph-level learning. The experiments show that GCKMs have at least competitive accuracy compared to GCNs.

**Strengths:**

1. The paper presents a framework GCKM for graph-based machine learning. As far as I am concerned, this is a novel framework. Compared to GCNs, GCKMs are easier to train, are guaranteed to obtain globally optimal solutions, and have strong generalization ability.
2. The authors provided generalization bound for GCKSVM, which justified the advantage of GCKSVM over KSVM.
3. The authors provided GCKM extensions to node clustering, graph classification, etc. They also provided fast feature transformation for GCKM.
4. The numerical evaluation are sufficient, which showed the proposed methods are at least as effective as GCNs.

**Weaknesses:**

Two minor issues are as follows.
1. In Section 3.3, the extension GCKPCA hasn’t been evaluated.
2. Some important results are in the supplement rather than the main paper. The authors may reorganize the paper and show the numerical justification for Theorem 1 and the experiment of graph-level learning in the main paper.

**Questions:**

1. In line 171, it is claimed “GCKM with Gaussian kernels can be easily generalized to multi-layer cases”. Could the authors provide a recursive or explicit formulation of deep GCKM? Is there a rule of thumb to determine the number of layers for deep GCKM?
2. In line 204, the formulation “...has tiny influences on the training...and the spectral norm of $[K_{ij}^{(L)}]_{i,j\in V}$...” is a little bit confusing. The matrix will become smaller if there are fewer support vectors and a smaller matrix usually has a smaller spectral norm.
3. In Eq. (18) or Eq. (16), the graph-level feature is obtained as the sum of nodes’ features. Is there any explanation for this operation? How about using the mean or min/max of nodes’ features as graph feature?
4. At the end of Section 3.4, I suggest the authors highlight the computational complexity in comparison to the implicit feature transformation method.
5. In the first column of Figure 3, why are the decision boundaries of SGC in the second and third plots are not nonlinear?

---

> ### Author Rebuttal · Authors · 2023-08-09
>
> **Reply to Weakness 1:**
> Thanks for your constructive advice. We have added a visualization experiment for GCKPCA, which is shown in the attached **PDF**.
> To be specific, we first map the node features to 2-D space by PCA, Graph-regularized PCA (GPCA) [1, 2] and GCKPCA, then further try to map them to 32-D space and leverage t-SNE to obtain the 2-D results.
> Figure 2 illustrates the comparison of mapping results of three methods. It can be observed that the figures of GCKPCA and GCKPCA + t-SNE both show the best separability between different classes, and GPCA performs slightly better than PCA.
>
> [1] Zhang, Zhenyue, and Keke Zhao. Low-rank matrix approximation with manifold regularization. TPAMI 2012.
>
> [2] Jiang, Bo, et al. Graph-Laplacian PCA: Closed-form solution and robustness. CVPR 2013.
>
> **Reply to Weakness 2:**
> Thanks for this helpful suggestion. We have adjusted our paper and will put these results on the additional page if accepted.
>
> **Reply to Question 1:**
> Thank you for raising this. To obtain the kernel matrix, we have the following recursive formulation:
> $$
> \begin{equation}
> {K}^{(l+1)}\_{i,j} = \exp\left(\frac{(\hat{\mathbf{A}}^{q})\_{i} \mathbf{K}^{(l)} (\hat{\mathbf{A}}^{q})^{\top}\_{i} - 2 (\hat{\mathbf{A}}^{q})_{i} \mathbf{K}^{(l)} (\hat{\mathbf{A}}^{q})^{\top}\_{j} + (\hat{\mathbf{A}}^{q})\_{j} \mathbf{K}^{(l)} (\hat{\mathbf{A}}^{q})^{\top}\_{j}}{2\sigma\_{l+1}^{2}}\right).
> \end{equation}
> $$
>
> where we define $\mathbf{K}^{(0)} = \mathbf{X} \mathbf{X}^{\top}$ in particular. For convenience, it can also be rewritten in matrix form:
>
> $$
> \begin{align}
> \begin{cases}
> \bar{\mathbf{K}}^{(l+1)} = \hat{\mathbf{A}}^{q} \mathbf{K}^{(l)} (\hat{\mathbf{A}}^{q})^{\top}, \\\\
> \mathbf{K}^{(l+1)}
> = \exp\left(-\frac{\mathbf{1}\_{n}^{\top} \mathbf{d}\_{\bar{\mathbf{K}}^{(l+1)}} + \mathbf{d}\_{\bar{\mathbf{K}}^{(l+1)}} \mathbf{1}\_{n} - 2 \bar{\mathbf{K}}^{(l+1)}}{2\sigma^{2}\_{l+1}}\right),
> \end{cases}
> \end{align}
> $$
>
> where $\mathbf{d}_ {\bar{\mathbf{K}}^{(l+1)}} = [\bar{K}_ {11}^{(l+1)}, \bar{K}_ {22}^{(l+1)}, \ldots, \bar{K}_ {nn}^{(l+1)}]$.
>
> We have also provided an experiment discussing the depth of GCKM in Figure 1.
> It shows that a 2-6 layer GCKM performs the best, attributed to which a deeper model requires more hyperparameters to be tuned.
> Besides, GCKM is not a neural network and thus benefits less from depth, and we can also increase the power of $\hat{\mathbf{A}}^{q}$ to reach more hops of neighborhood information in a layer.
>
> **Reply to Question 2:**
> Sorry for the misleading formulation. We meant that for a fixed number of support vectors, the graph convolution has a small influence on the spectral norm. But in reality, the graph convolution operation reduced the number of support vectors and the matrix become smaller, which leads to a smaller spectral norm.
>
> **Reply to Question 3:**
> Thank you for pointing out this. It is a widely adopted operator in graph-level tasks, which is called ReadOut function. All these min/max, mean, or sum of nodes' features are included in ReadOut functions, but [1] have proved that sum has a more powerful expressive ability over other ReadOut functions and we follow their setting to choose it. In practice, we also find that sum is better than other ReadOut functions.
>
> **Reply to Question 4:**
> Thanks for your helpful comment, we have revised our manuscript and highlighted this.
>
> **Reply to Question 5:**
> Actually, the decision boundaries of SGC are all nonlinear (because of the $\hat{\mathbf{A}}^q$) but approximately linear.
> The key idea of SGC is removing nonlinearities and collapsing weight matrices, thus the forward computation is simply formulated as
>
> $$
> \begin{equation}
> \mathbf{Z} = \hat{\mathbf{A}}^{q} \mathbf{X} \mathbf{W}.
> \end{equation}
> $$
>
> With a single learnable weight matrix and without activation functions, SGC's decision boundaries are all approximately linear, and the graph convolution provides slight nonlinearity to SGC.

---

> > ### Comment · Reviewer_PZjL · 2023-08-17
> >
> > Thank you for the detailed response and some of my concerns have been addressed. I tend to accept this paper.

---

> > > ### Author Response · Authors · 2023-08-17
> > > **Thanks for the feedback**
> > >
> > > We thank you very much for recoganizing our work.

---

### Official Review · Reviewer_YFNo · 2023-07-03

**Soundness:** 2 fair
**Presentation:** 2 fair
**Contribution:** 2 fair
**Rating:** 6
**Confidence:** 4

**Summary:**

This paper presents a kernel-based message-passing framework for graph convolutional networks called GCKM. The author demonstrated that GCKM is computationally efficient with stable performance on both node classification and graph classification tasks. Theoretical analyses of GCKM are provided.

**Strengths:**

1. GCKM modifies the general GNN by replacing the trainable parameters and non-linear activation functions with kernels (e.g., RBF Gaussian kernel). Classification is then done using well-established SVM. The proposed method significantly reduces the running time.
2. Provide theoretical analyses.


**Weaknesses:**

1. In section 3.4, the author claims that their method can be applied to very large graph datasets. What does “large” mean? In experiments, only small graph datasets (Cora, Citeseer, and PubMed) were used. How about relatively large datasets like Reddit or OGB datasets?
2. The value/insight of Theorem 1 is unclear. How can one use it to guide the design of GCKM models? How close is the theoretical bound to experimental observations? How can this address the challenges (like over-smoothing or improving model performances)?
3. Previous work [1] already conducts extensive exploration on applying kernel methods on GNNs, the author needs to discuss or compare their method with it.
4. The settings of the models used to produce Figures 1 and 2, e.g. the number of layers, and hidden dimensions, especially for the deep GCNs.
5. Abstract claims "GCKMs are guaranteed to obtain globally optimal solutions and have strong generalization ability and high interpretability." If true, should the GCKM produce the best results on all dataset? Interpretability need to be elaborated.

[1] Chen, Dexiong, Laurent Jacob, and Julien Mairal. "Convolutional kernel networks for graph-structured data." International Conference on Machine Learning. PMLR, 2020.


**Questions:**

1. What is the memory requirement of GCKM, when compared to other methods?

2. The over-smoothing problem in GCN or another type of GNN models is caused by stacking multiple message-passing layers, and eventually, every node shares similar embeddings. In Formula (5), a single GCKM layer is constructed by applying several times of graph convolution and using a pre-defined Gaussian kernel function as the ReadOut function. How does this architecture address the over-smoothing problem?



**Limitations:**

Not sure if the choice/design of kernel may cause any biases.

---

> ### Author Rebuttal · Authors · 2023-08-09
>
> **Reply to Weakness 1:**
> Thanks for your valuable suggestions.
> We added a comparison experiment on OGB dataset Arxiv with about 160k nodes. Table 1 (in the attached **PDF**) demonstrates that GCKM is still competitive with these GNNs on large-scale dataset. Although the complexity of GCKM is quadratic, we provide an efficient variant GCKM-E in Section 3.4 that leverages the random Fourier feature to explicitly derive a low-dimensional output instead of a kernel matrix. With this output, we can apply fast linear methods (e.g. linear SVM) for downstream tasks, which is still more efficient than GNNS needing a lengthy training process.
>
> **Reply to Weakness 2:**
> The theorem showed that the generalization error bound is linear with $\vert\mathcal{V}\vert$ and $\big\\|[K^{(L)}\_{ij}]\_{i,j\in\mathcal{V}}\big\\|\_{\text{spec}}$. We showed in Table 4 of Appendix (or an intercepted part in **Table 3 of the global rebuttal PDF file**) that the graph convolution can reduce both $\vert\mathcal{V}\vert$ and $\big\\|[K^{(L)}\_{ij}]\_{i,j\in\mathcal{V}}\big\\|\_{\text{spec}}$, and then lead to a tighter error bound. A tighter generalization error bound means that the test error is potentially smaller. In other words, graph convolution improves the model performance.
>
> **Reply to Weakness 3:**
> Many thanks for pointing out the reference [1] (GCKN). It is indeed an interesting paper. We have revised our manuscript and discussed this work. Here we mainly state the differences and our unique contributions:
>
> 1. Motivated by the previous studies of "graph kernel" (computing the similarity between entire graphs) and GNNs, GCKN [1] focuses on graph-level tasks and aims to connect graph kernel and GNNs. Differently, our GCKM is motivated by the unsatisfactory performance of deep GNNs and the possibility of classical kernel machine learning.  GCKM is built upon kernels rather than neural networks. In other words, GCKM is a general framework that can be applied to both node-level tasks and graph-level tasks. It is parallel to GNNs.
> 2. GCKN constructs a neural network via employing the graph kernels and shows decent performance on graph-level tasks, while GCKMs are a series of graph convolutional kernel based machine learning approaches and perform well on several downstream tasks. Nevertheless, due to the path kernel, GCKN can be time-consuming when the path lengths is long.
> GCKM has advantages like a globally optimal solution, faster computation, higher interpretability, and a stronger theoretical guarantee.
> 3. We propose an efficient variant GCKM-E to explicitly compute the node representation and can also be easily extended.
> 4. We provide theoretical and empirical analyses of the graph's influence on the test error bound.
> 5. We now demonstrate that GCKM can perform well on large-scale OGB dataset and can be deeper and alleviate the so-called over-smoothing issue.
>
> [1] Convolutional kernel networks for graph-structured data. ICML 2020.
>
> **Reply to Weakness 4:**
> Thank you for this valuable advice. Figure 2 is the flowchart of GCKM, maybe you mean Figure 3? For Figure 1, we set the hidden dimensions as $32$ for all hidden layers of GCN, APPNP, and JKNet, while SGC only has a learnable matrix $\mathbf{W} \in \mathbb{R}^{n\times c}$ where $n$ is the number of nodes and $c$ is the number of classes, and learning rate is selected in $\{ 1 \times 10^{-2}, 1 \times 10^{-3}, 1 \times 10^{-4}\}$ and weight decay is selected in $\{ 5 \times 10^{-4}, 5 \times 10^{-5}, 5 \times 10^{-6}\}$.
> For Figure 3, the numbers of layers are $2$ and $8$ for GCN and APPNP respectively, the hidden dimensions are $32$, and the learning rate is set as $1 \times 10^{-3}$ and weight decay is $5 \times 10^{-5}$ for all models.
>
> **Reply to Weakness 5:**
> Thank you for raising these. Actually, we claimed the following opinions and corresponding reasons:
> 1. The model can achieve a globally optimal solution because the optimization problem is convex.
> 2. Theorem 1 and the experiments demonstrate good generalization ability.
> 3. The model has higher interpretability because the support vectors form the decision boundary.
>
> These three statements about GCKM are the potential reasons for GCKMs' good performance. They do not mean that GCKMs should produce the best results on all datasets in comparison to other methods. Here the optimality means in terms of optimization, GCKM can obtain its optimal solution given the current architecture and hyperparameters on a specific dataset.
>
> **Reply to Question 1:**
> The space complexity of GCKM and GCKM-E are $\mathcal{O}(n^2)$ and $\mathcal{O}((n+d)m)$ respectively, while a typical GNN, like GCN, requires $\mathcal{O}((n+d')m)$, if the edges are sparse, where $n$ is the number of nodes, $m$, is the number of input feature dimension, $d$ and $d'$ are the hidden dimensions of GCKM-E and GNN respectively. Although GCKM has high memory requirement when the graph is large, we can use GCKM-E paired with a linear method (e.g. linear SVM) to reduce the complexity.
>
> **Reply to Question 2:**
> Thanks for pointing out this.
> The ReadOut function is only adopted in the graph-level variant, and the over-smoothing problem is mainly discussed in the context of node-level tasks, so we analyze this phenomenon on the node classification task.
> We have supplemented an experiment on this issue in **Figure 1 of the global rebuttal PDF file**, which revealed that **GCKM can be deeper and alleviate the over-smoothing issue**.
> Due to the limitation of characters, **please refer to the Reply to Question 1 in response to Reviewer dP1e for detailed analyses.**
>
> **Reply to Limitation 1:**
> Thank you for pointing out this limitation. We have provided a complementary experiment on various kernel functions in **Table 2 of the global rebuttal PDF file**, including 2nd order polynomial kernel, sigmoid kernel, and Laplacian kernel. All the kernel functions show decent performance and the Gaussian kernel performs the best.

---

> > ### Comment · Reviewer_YFNo · 2023-08-18
> >
> > Thanks authors for providing additional information including experiments. I raised my score.

---

> > > ### Author Response · Authors · 2023-08-18
> > > **Many thanks**
> > >
> > > We sincerely thank you for recognizing our work and increasing the score.

---

### Official Review · Reviewer_nd8y · 2023-07-04

**Soundness:** 4 excellent
**Presentation:** 3 good
**Contribution:** 3 good
**Rating:** 6
**Confidence:** 4

**Summary:**

This paper proposes a new support vector machine approach for graph learning called graph convolutional kernel machine. GCKM combines traditional kernel functions with graph convolution, and shows good performance in both node- and graph-level tasks. Generalization bound is also provided for the approach.

**Strengths:**

1. The proposed GCKM is simple yet effective. It is impressive that SVM-based approach can perform on par with deep GNN models in both node- and graph-level tasks.
2. Generalization bound is provided for the proposed GCKM, which partially explains how graph structure in the construction of kernel function benefits generalization performance.
3. Many variants are developed for applications in different scenarios.

**Weaknesses:**

1. Though many datasets and tasks are considered in the paper, all datasets are quite small with the largest one consisting of only 20k nodes. While the paper claims GCKM can be "extended to large-scale data” by using low-rank approximation tricks, no experimental results on large datasets are provided. Additionally, I would also suggest the authors put some results in supplementary experiments to the main text as cora/citeseer/pubmed are somewhat outdated and insufficient to reflect the effectiveness of the proposed approach.
2. The explanation of how the graph structure affects the generalization of GCKM is somewhat vague. Specifically, I am still confused of why “graph structure significantly improved the quality of the kernel matrix” and how it affects the generalization bound.
3. Literature coverage could be improved. Some prior work have also analyzed how graph structure in the construction of kernel function affects generalization in graph [1] and node [2] tasks, and some others adopted kernel methods for graph learning e.g. [3,4], which are related to this work.
4. As the authors stated in conclusion, we did not systematically test other kernel functions.

[1] Graph Neural Tangent Kernel: Fusing Graph Neural Networks with Graph Kernels, NeurIPS 2019
[2] Graph Neural Networks are Inherently Good Generalizers Insights by Bridging GNNs and MLPs, ICLR 2023
[3] Convolutional Kernel Networks for Graph-Structured Data, ICML 2020
[4] KerGNNs: Interpretable Graph Neural Networks with Graph Kernels, AAAI 2022

**Questions:**

1. How does the GCKM perform on larger datasets, e.g. open graph benchmark? Such the complexity of GCKM is quadratic, is it still more efficient than GNNs in those larger datasets?
2. Why “graph structure significantly improved the quality of the kernel matrix” and how it affects the generalization bound?

**Limitations:**

See limitations in weakness section.

---

> ### Author Rebuttal · Authors · 2023-08-09
>
> **Reply to Weakness 1/Question 1:**
> Thanks for your constructive suggestions.
> We have added a comparison experiment on the OGB dataset Arxiv with about 160k nodes. Table 1 (see the attached **PDF**) demonstrates that GCKM is still competitive with these GNNs on large-scale datasets. Although the complexity of GCKM is quadratic, we provide an efficient variant GCKM-E in Section 3.4 that leverages the random Fourier feature to explicitly derive a low-dimensional output instead of a kernel matrix. With this output, we can apply fast linear methods (e.g. linear SVM) for downstream tasks, which is still more efficient than GNNS needing a lengthy training process.
>
> **Reply to Weakness 2/Question 2:**
> Thank you for the valuable comment. Theorem 1 theoretically demonstrated the connections between the generalization bound and graph structure, and experiments further provide the evidence.
> The results are recorded in Table 4 of Appendix, here we intercept a part of the table in **Table 3 of the global rebuttal PDF file**.
>
> We considered the following three cases for a comprehensive comparison.
> 1. **non-graph-convolution**  We replaced the affinity matrix $\hat{\mathbf{A}}$ in GCKM with an identity matrix $\mathbf{I}_n$, which means the graph structure is not used.
> 2. **strongly connected graph**  $\ \hat{\mathbf{A}}$ in GCKM is replaced by $\mathbf{I}\_{n}-\mathbf{1}\_{n \times n}$, which means every node is connected with all other nodes.
> 3. **normal GCKM**
>
> With fixed $\lambda$, graph convolution significantly reduces the number of support vectors and spectral norm on the kernel matrix of support vectors.
> According to Theorem 1, the upper bound (linear with $\vert\mathcal{V}\vert$ and $\big\\|[K^{(L)}\_{ij}]\_{i,j\in\mathcal{V}}\big\\|\_{\text{spec}}$) of test error can be reduced compared to using a non-graph-convolution kernel.
> Thus, we can conclude that the graph structure significantly improved the quality of the kernel matrix.
>
> **Reply to Weakness 3:**
> Thank you for raising this problem. We have revised our manuscript, discussing the differences and citing these related research.
>
> **Reply to Weakness 4:**
> Thank you for pointing out this limitation. We have provided a complementary experiment on various kernel functions in **Table 2 of the global rebuttal PDF file**, including 2nd order polynomial kernel, sigmoid kernel, and Laplacian kernel.
> All the kernel functions show decent performance and the Gaussian kernel performs the best.

---

> > ### Comment · Reviewer_nd8y · 2023-08-18
> >
> > Thank you for your response. Some of my concerns are addressed, and I find the extension to large graphs particularly valuable. Moreover, while it may not be a central concern, I am still confused why "graph convolution significantly reduces the number of support vectors". Is this claim provable or just an empirical observation? If this claim lacks support from a theorem, it would be beneficial to clarify this point in the main text. Overall, I appreciate the simplicity and effectiveness of proposed approach, and will keep the score as 6.

---

> > > ### Author Response · Authors · 2023-08-18
> > > **Authors' feedback**
> > >
> > > Thank you very much for the comment. "graph convolution significantly reduces the number of support vectors" is an empirical observation. Since the number of support vectors is data-dependent, we cannot prove it theoretically unless making assumptions about the data.
> > >
> > > We here prove the claim theoretically based on the following assumption:
> > >
> > > **Assumption:** Convolution with graph $G$ increases the inner product between the kernel feature maps of samples in the same class and reduces or does not change the inner product between the kernel feature maps of samples in different classes.
> > >
> > > This is a reasonable assumption because a useful graph should make the samples from different classes more distinguishable or at least make the samples from the same class more similar.
> > >
> > > Let $\varphi$ and $\varphi_G$ be the kernel feature map without and with graph convolution respectively.
> > > Recall the Lagrangian dual problem:
> > >
> > > \begin{equation}
> > >  \qquad\mathop{\text {max}}_ {\mathbf{c}} ~~\sum_{i=1}^n c_i-\frac{1}{2} \sum_ {i=1}^n \sum_ {j=1}^n c_ ic_ jy_ iy_ j{\varphi(\mathbf{x}_ i)}^{\top} {\varphi(\mathbf{x}_ j)}\qquad
> > >  \text {s.t.} \sum_{i=1}^n c_ i y_ i=0, ~0 \leq c_ i \leq \frac{\lambda}{n}.
> > > \end{equation}
> > > For convenience, we let $\mathcal{L}(\mathbf{c}):=\frac{1}{2} \sum_{i=1}^n \sum_{j=1}^n c_ic_jq_{ij}-\sum_{i=1}^n c_i$, where $q_ {ij}=y_ iy_ j{\varphi(\mathbf{x}_ i)}^{\top} {\varphi(\mathbf{x}_ j)}$. Then the problem is equivalent to
> > >
> > > \begin{equation}
> > >  \qquad\mathop{\text {min}}_ {\mathbf{c}} ~~\mathcal{L}(\mathbf{c})\qquad
> > >  \text {s.t.} \sum_{i=1}^n c_ i y_ i=0, ~0 \leq c_ i \leq \frac{\lambda}{n}.
> > > \end{equation}
> > >
> > > Similarly, for the case of using the graph $G$, we let
> > > $\mathcal{L}_ {G}(\mathbf{c}):=\frac{1}{2} \sum_ {i=1}^n \sum_ {j=1}^n c_ ic_ jq_ {ij}^{G}-\sum_ {i=1}^n c_ i$, where $q_{ij}^{G}=y_iy_j{\varphi_G(\mathbf{x}_i)}^{\top} {\varphi_G(\mathbf{x}_j)}$.
> > >
> > > According to the previous assumption, we have:
> > >  * if samples $i$ and $j$ are in the same class, $\varphi_G(\mathbf{x}_i)^\top\varphi_G(\mathbf{x}_j)>\varphi(\mathbf{x}_i)^\top\varphi(\mathbf{x}_j)$ and $y_iy_j=1$;
> > >  * if samples $i$ and $j$ are in different classes, $\varphi_ G(\mathbf{x}_ i)^\top\varphi_ G(\mathbf{x}_ j)\leq\varphi(\mathbf{x}_ i)^\top\varphi(\mathbf{x}_ j)$ and $y_ iy_ j=-1$.
> > >
> > > Therefore, the following inequality holds:
> > > $$\qquad q_{ij}^G=q_{ij}+\epsilon_{ij}, \text{. where } \epsilon_{ij}\geq 0~\forall (i,j)\in[n]\times[n].$$
> > >
> > >
> > > For convenience, let $\bar{\epsilon}_ i=\min_ {j}\epsilon_ {ij}$ and $\tilde{\epsilon}=\min_ {i}\bar{\epsilon}_ {i}$.
> > > We have
> > > \begin{equation}
> > > \begin{aligned}
> > >     \mathcal{L}_ {G}(\mathbf{c}):=&\frac{1}{2} \sum_ {i=1}^n \sum_ {j=1}^n c_ ic_ j(q_ {ij}+\epsilon_ {ij})-\sum_ {i=1}^n c_ i\\\\
> > >     \geq&\frac{1}{2} \sum_ {i=1}^n \sum_ {j=1}^n c_ ic_ jq_ {ij}+\frac{1}{2} \sum_ {i=1}^n \bar{\epsilon}_ i\sum_ {j=1}^n c_ ic_ j-\sum_ {i=1}^n c_ i\\\\
> > >     =&\mathcal{L}(\mathbf{c})+\frac{1}{2} \sum_ {i=1}^n \tilde{\epsilon}_ ic_ i\sum_ {j=1}^n c_ j\\\\
> > >     \geq&\mathcal{L}(\mathbf{c})+\frac{1}{2}\left(\tilde{\epsilon}\Vert \mathbf{c}\Vert_ 2^2+\sum_ {i=1}^n\bar{\epsilon}_ ic_ i\Vert\mathbf{c}_ {/i}\Vert_ 1\right),
> > > \end{aligned}
> > > \end{equation}
> > > where $\mathbf{c}_ {/i}=[c_ 1,\ldots,c_ {i-1},c_ {i+1},\ldots,c_ n]^\top$.
> > > It is known that the $\ell_1$-norm $\Vert\cdot\Vert_1$ is a convex relaxation of the $\ell_0$-norm $\Vert\cdot\Vert_0$, i.e., the number of nonzero elements in a vector. Denote $\mathcal{R}(\mathbf{c}):=\tilde{\epsilon}\Vert \mathbf{c}\Vert_ 2^2+\sum_ {i=1}^n\bar{\epsilon}_ ic_ i\Vert\mathbf{c}_ {/i}\Vert_ 1$. We see $\mathcal{R}(\mathbf{c})$ is very similar to the elastic net regularization and is able to induce sparsity. Actually, if we let $\kappa=\min_ i\Vert\mathbf{c}_ {/i}\Vert_ 1$, we have $\mathcal{R}(\mathbf{c})\geq \tilde{\epsilon}\Vert \mathbf{c}\Vert_ 2^2+\kappa\sum_ {i=1}^n\bar{\epsilon}_ ic_ i=\tilde{\epsilon}\Vert \mathbf{c}\Vert_ 2^2+\kappa\Vert\text{diag}(\bar{\boldsymbol{\epsilon}})\mathbf{c}\Vert_1$, where the second term is a weighted $\ell_1$-norm and also induces sparisty.
> > >
> > > Therefore, the graph convolution introduces an additional sparse regularization term $\mathcal{R}(\mathbf{c})$, which will make $\mathbf{c}$ sparser, or in other words, reduce the number of support vectors.
> > >
> > > We will form a proposition using the above result and add it to the paper.
> > > We hope that this analysis could make you be more confident about our work and increase the score if possible.

---

> > > > ### Comment · Reviewer_nd8y · 2023-08-19
> > > >
> > > > Thank you for your timely clarification. I am quite satisfied with the new analysis. I will raise the 'soundness' rating to 4, while keeping the overall rating as 6. I am confident the paper will be accepted given the other good scores.

---

> > > > > ### Author Response · Authors · 2023-08-19
> > > > > **Authors‘ comment**
> > > > >
> > > > > Thank you very much for encouraging us.

---

### Official Review · Reviewer_dP1e · 2023-07-07

**Soundness:** 3 good
**Presentation:** 3 good
**Contribution:** 3 good
**Rating:** 6
**Confidence:** 3

**Summary:**

This paper introduces a novel approach called the Graph Convolutional Kernel Machine (GCKM) for graph learning. Unlike other neural network-based frameworks, GCKM employs a graph kernel to replace the neighbor aggregation step, without any learnable parameters.
Then the author build the features based on the graph kernel and then use SVM for classification tasks.

In general, I find the paper to be good and interesting, although it could benefit from additional experiments. I am somewhat surprised by the better performance of GCKSVM compared to GAT, considering that GAT defines the similarity of node features using learnable attention weights.

**Strengths:**

1. Well written and clear explained
2. good visualization (Figure 3) of the potential benefit of kernel methods: the Kernel methods offer greater interpretability compared to neural networks and provide stronger generalization guarantees.  In contrast to SGC, which functions as a linear classifier based on the final node representation, GCKSVM serves as a nonlinear classifier. Therefore, it is anticipated to outperform SGC in cases where the data is not linearly separable.
3. Faster running time.

**Weaknesses:**

1. The performance is relatively underwhelming. Table 2 reveals that GCKSC achieves the best results in only 3 cases and the second-best results in 2 cases, whereas S3 GC outperforms with the best results in 3 cases and the second-best results in 3 cases.
2. The performance reported in this paper appears to differ significantly from the results presented in the APPNP paper (https://arxiv.org/pdf/1810.05997v6.pdf). In the APPNP paper, they reported 85% accuracy on Cora and 75% accuracy on Citeseer. The substantial difference in performance raises questions about potential differences in the experimental settings between the two papers. If there are indeed differences in the experimental settings, it would be important to assess whether the conclusions drawn in this paper still hold when utilizing the experimental settings from the APPNP paper.

**Questions:**

The author discusses the issue of over-smoothing, and it raises the question of whether over-smoothing would occur when employing GCKSC with more layers. Specifically, is there a significant drop in performance when utilizing GCKSC with more than 2 layers?

---

> ### Author Rebuttal · Authors · 2023-08-09
>
> **Reply to Weakness 1:**
> Based on the analysis in our paper, GCKM aims to further build a simple paradigm for graph-oriented tasks.
> It can be viewed as a further simplified baseline and its main competitors are GCN, GAE, VAGE and other simplified models,
> but GCKMs have shown outstanding performance over not only baselines but also some SOTA methods.
> And we note that all the competitors are deep learning based methods and GCKMs are a series of kernel-based traditional methods.
> Thus, besides the competitive performance, our models also have the following advantages:
> 1. The model can achieve a **globally optimal solution** because the optimization problem is convex.
> 2. The computation is **faster** because the model does not involve forward and backward propagation in the training and inference stages.
> 3. The model has **higher interpretability** (e.g. the support vectors of GCKSVM form the decision boundary).
> 4. The model has a **stronger theoretical guarantee** (the generalization error bound is almost tight).
>
> In contrast, the optimizations of GNNs are nonconvex and it is very difficult or even impossible to obtain globally optimal solutions. Moreover, it is well known that the generalization bounds of neural networks are usually exponentially dependent on the network depth [1]. Last but not least, the decision process of neural networks has much lower interpretability.
>
> [1] Norm-based capacity control in neural networks. COLT 2015.
>
> **Reply to Weakness 2:** Thank you for pointing out this. Although the dataset names are the same, they are actually **different** datasets.
> The original paper of APPNP adopted Citeseer with 2,110 nodes and Cora-ML with 2,810 nodes, while we use Citeseer with 3,327 nodes and Cora with 2,708 nodes following settings of vanilla GCN [1] (please refer to Table 1 in Appendix of our paper and Table 1 in APPNP paper), which is widely adopted in node classification methods [2, 3, 4]. There are also some recent works [3, 6, 7] that evaluated APPNP under this setting, and comparing with their results, we believe ours are fair and reasonable.
>
> [1] Semi-supervised classification with graph convolutional networks. ICLR 2017.
>
> [2] Simplifying graph convolutional networks. ICML 2019.
>
> [3] Dissecting the diffusion process in linear graph convolutional networks. NeurIPS 2021.
>
> [4] Beyond low-frequency information in graph convolutional networks. AAAI 2021.
>
> [5] Dropmessage: Unifying random dropping for graph neural networks. AAAI 2023.
>
> [6] Node-wise Diffusion for Scalable Graph Learning. WWW 2023.
>
> [7] Elastic graph neural networks. ICML 2021.
>
> **Reply to Question 1:** Thanks for your insightful comment. We have supplemented an experiment (**Figure 1 of the global rebuttal PDF file**) on this issue, which revealed that **GCKM can be deeper and alleviate the over-smoothing issue**. To be specific, two situations are considered in this experiment:
>
> 1. GCKM with fixed 2 layers and varied hops of neighbors per aggregation ($q$ in Eq. (5))
>    $$
>    \begin{equation}
>    \mathbf{H} = \phi_{(1)}( \hat{\mathbf{A}}^{q} \phi_{(0)}(\hat{\mathbf{A}}^{q} \mathbf{X})).
>    \end{equation}
>    $$
>
> 2. GCKM with fixed 2 hops of neighbors per aggregation and varied layers
>    $$
>    \begin{equation}
>    \mathbf{H} = \phi_{(l)}( \hat{\mathbf{A}}^{2} \cdots \phi_{(0)}(\hat{\mathbf{A}}^{2} \mathbf{X})).
>    \end{equation}
>    $$
>
> From **Figure 1 in the global rebuttal PDF file**, we have the following observations:
>
> 1. Deep GCKM performs more stably than deep GCN.
> 2. GCKM's performance first improves then slightly decreases with increasing layers/hops
> 3. Particularly, GCKM with fixed layers performs better and decreases less.
>
> It is known that over-smoothing is caused by the aggregation step, that is, multiplying $\hat{\mathbf{A}}$ makes the representations of different nodes more and more indistinguishable. However, the analysis of over-smoothing does not consider activation functions and learnable weights, and it theoretically exists when the power of $\hat{\mathbf{A}}$ tends to infinity [1], which does not match the fact that GCN collapses with only 8 layers. Recent studies [2, 3] have pointed out that over-smoothing problem might be an artifact of theoretical analysis and the failure of deep GNNs may not only cause by the over-smoothing issue in the aggregation/message-passing step.
>
> Although GCKM and GCN share a similar aggregation step, the main difference between them is the transformation step, namely, GCN uses a linear layer to conduct explicit dimension reduction while GCKM employs implicit high-dimensional feature mapping.
> GCKM implicitly maps node features to a high-dimensional (even infinite) space after aggregation, and there may exist an appropriate space where node representation can be distinguished. In contrast, recent studies [3] have found that the node representations processed by deep GCN would collapse to be low-rank and lose expressive power. The appropriate space can be found by tuning the hyperparameters of GCKM, however, the number of hyperparameters increases when building a deeper GCKM and makes it time-consuming to search for this space. That may be the reason why GCKM decreases slightly with too many layers and why fixing the number of layers improves the performance.
>
> [1] Deeper insights into graph convolutional networks for semi-supervised learning. AAAI 2018.
>
> [2] On provable benefits of depth in training graph convolutional networks. NeurIPS 2021.
>
> [3] Contranorm: A contrastive learning perspective on oversmoothing and beyond. ICLR2023.

---

> > ### Comment · Reviewer_dP1e · 2023-08-19
> >
> > thank you for your response, I now raise to 6.

---

> > > ### Author Response · Authors · 2023-08-19
> > > **Thanks for the feedback**
> > >
> > > We greatly appreciate your comments and recognition.

---

### Author Rebuttal · Authors · 2023-08-09

We would like to thank the Senior Area Chairs/Area Chairs and all the Reviewers for handling our paper and providing constructive comments.
We have systematically and carefully replied to all the comments and revised our work based on comments from the reviewers.
In addition, we have attached a PDF file including four experiments to support our response, and all these source codes can be provided if needed.
The following is the summary of mainly focused problems and our response:
1. **Deep GCKM and over-smoothing issue.** We have conducted experiments (**Figure 1 of the PDF file**) considering two situations and showed that GCKM can be deeper and alleviate the over-smoothing issue. Besides, we analyzed the possible reasons in the response.
2. **Performance and efficiency on large-scale datasets.** We have provided results on the OGB dataset Arixv with over 160k nodes (**Table 1 of the PDF file**), where GCKM is still competitive with other baseline and SOTA methods.
3. **Explanation of Theorem 1.** We have further explained Theorem 1 and put the corresponding experimental results in **Table 3 of the PDF file** (full table in Appendix). Theoretical and empirical results revealed that graph convolution in GCKM can improve the generalization bound.
4. **Experiments of GCKPCA.** The results are in Figure 2 of the attached PDF file. GCKPCA outperformed PCA and GPCA.

Besides these, we also evaluated GCKM with different kernel functions (**recorded in Table 2 of the PDF file**) and also further elaborated on the experimental settings, discussion with related work, the effectiveness of GCKM, etc.

Finally, many thanks to the positive assessments that encouraged us a lot:
1. R1: "Well written and clear explained; good visualization."
2. R2: "The proposed GCKM is simple yet effective; it is impressive."
3. R4: "This is a novel framework."

---

### Decision · Program_Chairs · 2023-09-21

**Decision:**

Accept (poster)

**Comment:**

This paper introduces a new approach called the Graph Convolutional Kernel Machine (GCKM) for graph-based machine learning, which are built upon kernel functions integrated with graph convolution. Overall, the reviewers found the paper to be well-written, and they found the results to be both interesting and significant. However, they suggested that additional experiments could enhance the paper's quality. The authors are encouraged to consider the reviewers' feedback during the revision process.